# Revisit Non-parametric Two-sample Testing as a Semi-supervised Learning Problem

## Abstract

Learning effective data representations is crucial in answering if two samples X and Y are from the same distribution (a.k.a. the non-parametric two-sample testing problem), which can be categorized into: i) learning *discriminative representations* (DRs) that distinguish between two samples in a *supervised-learning* paradigm, and ii) learning *inherent representations* (IRs) focusing on data's inherent features in an *unsupervised-learning* paradigm. However, both paradigms have issues: learning DRs reduces the data points available for the two-sample testing phase, and learning purely IRs misses discriminative cues. To mitigate both issues, we propose a novel perspective to consider non-parametric two-sample testing as a *semi-supervised learning* (SSL) problem, introducing the *SSL-based Classifier Two-Sample Test* (SSL-C2ST) framework. While a straightforward implementation of SSL-C2ST might directly use existing *state-of-the-art* (SOTA) SSL methods to train a classifier with labeled data (with sample indexes X or Y) and unlabeled data (the remaining ones in the two samples), conventional two-sample testing data often exhibits substantial overlap between samples and violates SSL methods' assumptions, resulting in low test power. Therefore, we propose a two-step approach: first, learn IRs using all data, then fine-tune IRs with only labelled data to learn DRs, which can both utilize information from whole dataset and adapt the discriminative power to the given data. Extensive experiments and theoretical analysis demonstrate that SSL-C2ST outperforms traditional C2ST by effectively leveraging unlabeled data. We also offer a stronger empirically designed test achieving the SOTA performance in many two-sample testing datasets.

## 1 Introduction

Two-sample tests aim to solve the problem of "Whether two samples are drawn from the same distribution?". Classical two-sample tests, including *t*-tests which test the empirical mean differences between two samples, often need to assume that samples are drawn from specific distributions (e.g., Gaussian distributions with the same variance). To alleviate the strict assumptions, non-parametric two-sample tests are proposed to solve the problem only based on observed data (Gretton et al., 2012a;b; Heller & Heller, 2016; Székely & Rizzo, 2013; Jitkrittum et al., 2016; Chen & Friedman, 2017; Ghoshdastidar et al., 2017; Lopez-Paz & Oquab, 2018b; Ramdas et al., 2017; Sutherland et al., 2017; Gao et al., 2018; Ghoshdastidar & von Luxburg, 2018; Lerasle et al., 2019; Liu et al., 2020; Kirchler et al., 2020; Kübler et al., 2020; Cheng & Xie, 2021; Kübler et al., 2022; Kübler et al., 2022; Liu et al., 2021; Deka & Sutherland, 2023; Bonnier et al., 2023).

For example, the *Kolmogorov-Smirnov* (K-S) test is designed to compare the cumulative distribution functions derived from two samples, but it can only be effective in extremely low-dimensional data (Kolmogorov, 1933; Smirnov, 1948). The *maximum mean discrepancy* (MMD) test adopts the kernel mean embedding of distribution and uses it to measure the discrepancy between two distributions Gretton et al. (2012a). The statistics used in these non-parametric two-sample tests are also widely adopted in many other fields, such as domain adaptation, causal discovery, generative modeling, adversarial learning, and more (Gong et al., 2016; Bińkowski et al., 2018; Stojanov et al., 2019; Cano & Krawczyk, 2020; Oneto et al., 2020; Gao et al., 2021; Fang et al., 2021b; Zhong et al., 2021; Fang et al., 2021a; Song et al., 2021a; Tahmasbi et al., 2021; Taskesen et al., 2021; Bergamin et al., 2022).

To improve the test power of non-parametric two-sample tests in practical applications, recent studies have shown that learning good data representations is crucial before performing two-sample testing (Kirchler et al., 2020; Liu et al., 2020; 2021; Gao et al., 2021; Bergamin et al., 2022). For example, Kirchler et al. (2020) directly use a pre-trained feature extractor to extract features of two samples and find it is useful to increase the test power during the testing. Meanwhile, Liu et al. (2020) propose a unified learning paradigm to learn deep-net representations of data via maximizing the test power of MMD and show that the learned representations can help capture the difference between two samples. Recently, Biggs et al. (2023) point out that, after discarding the sample information (namely, we do not know which sample the data belongs to), learning representations from whole samples will not influence the type I error of permutation-based testing methods, which further justifies the correctness of learning good representations for testing.

**Two learning paradigms and their issues.** There are two main data representation learning paradigms in the two-sample testing field: 1) the supervised paradigm; and 2) the unsupervised paradigm (see Figure 1). In paradigm 1), we first split samples into training and testing sets, then learn a representation extractor to obtain two samples' *discriminative representations* (DRs) (Sutherland et al., 2017; Lopez-Paz & Oquab, 2018b; Liu et al., 2020; 2021). In paradigm 2), we can learn a representation extractor based on data from the whole samples after discarding the sample information (Biggs et al., 2023). For example, unsupervised learning can be used to learn *inherent representations* (IRs) of samples (Biggs et al., 2023).

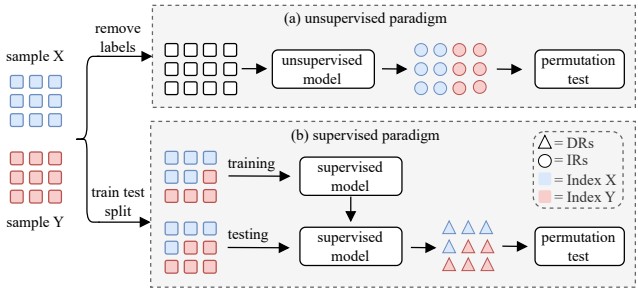

Figure 1: Visualisation of two learning paradigms. Blue color represents data with sample index X, red color represents data with sample index Y, and transparent represents data without sample index information. The square represents original input samples, the circle represents the *inherent representations* (IRs) learned from unsupervised model, and the triangle represents the *discriminative representations* (DRs) learned from supervised model.

Although the supervised paradigm has obtained success in many fields (Gao et al., 2021; Bergamin et al., 2022), we have to use part of samples to train a good classifier (Lopez-Paz & Oquab, 2018b) or a kernel function (Liu et al., 2020), which will cause fewer samples are used in the final testing procedure. Namely, the procedure of splitting samples into training and testing sets will naturally lower the test power. There *has to be a trade-off* between the extra power provided by the learned functions/kernels and the sacrificed power due to the sample-splitting procedure. For example, Biggs et al. (2023) recently reveal that combining several pre-defined kernels on the whole samples can provide higher test power compared to deep-kernel MMD test (Liu et al., 2020) on some datasets, indicating that, in some cases, the sacrificed power might be higher than the extra power provided by learned functions or kernels.

In the unsupervised paradigm, researchers try to develop testing methods that do not need the data-splitting procedure. To avoid sacrificing power from the data-splitting procedure, Kübler et al. (2020) propose a new testing method based on the linear-time estimator of MMD and the selective inference framework. Because Kübler et al. (2020) use a linear-time estimator of MMD, there is a test-power reduction compared to the U-statistic or V-statistic of MMD. Then, Schrab et al. (2023) and Biggs et al. (2023) propose new ways to combine several kernels in a *given candidate set* and perform the two-sample testing directly on the whole samples. Empirical experiments support that their newly proposed statistic, MMD-FUSE, enjoys even higher test power than the most effective method in the first paradigm given a good candidate set. However, there is still an open question in this paradigm: can we always expect a relatively good kernel in the candidate set for any given two samples?

**Motivation.** Based on the development of the two-sample testing methods reviewed above, it can be seen that both the supervised paradigm and unsupervised paradigm have their own issues. For the supervised paradigm, we have to use a relatively large amount of data to ensure that we can learn a good function or kernel, resulting in a possibly higher sacrificed power. For the unsupervised paradigm, a good candidate set is key but we do not have supervision to find such a candidate set.

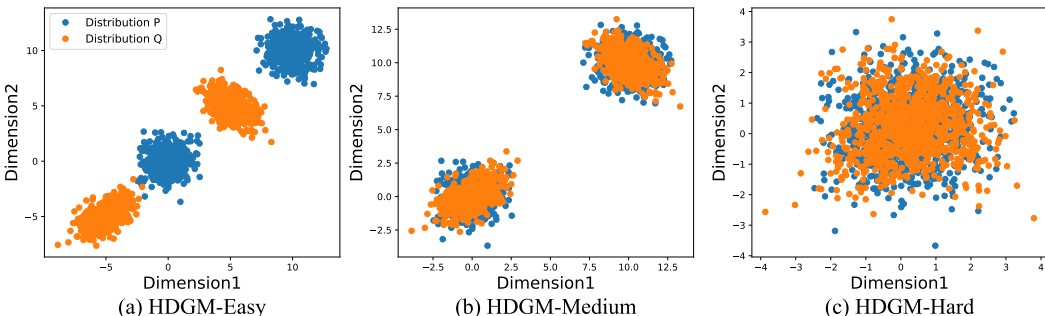

Figure 2: Visualisation of first two dimensions of samples for different levels of the HDGM dataset whose dimension is 10. For the HDGM-Easy and HDGM-Medium, the cluster mean difference $\Delta_\mu$ within the same distribution is 10, while for the HDGM-Hard, $\Delta_\mu$ is 0.5. For the HDGM-Easy, the distribution mean difference $\Delta_q$ between $\mathbb{P}$ and $\mathbb{Q}$ is 5, while for HDGM-Medium and HDGM-Hard, $\Delta_q$ is 0. Other setting of how to generate HDGM dataset is described in Appendix C.4

Thus, to obtain a better two-sample testing method, we might aim to *reduce the requirement of a large training set* in the supervised paradigm or to *provide supervision to find a good kernel* in the unsupervised paradigm. The first aim is quite similar to the advantage that *semi-supervised learning* (SSL) can bring in classification: Given many unlabeled data, SSL methods can help us obtain a good classifier even the training set is *small* (Balcan & Blum, 2010).

**Our contributions.** In this paper, we revisit the non-parametric two-sample testing as an SSL problem and propose an *SSL-based classifier two-sample test* (SSL-C2ST). SSL-C2ST extends the state-of-the-art (SOTA) two-sample testing method C2ST by incorporating SSL methods.

In our experiments, we firstly implemented several SOTA SSL methods, depending on different SSL frameworks, such as *consistency regularization* (CR) (Xie et al., 2020), *pseudo labeling* (PL) (Lee et al., 2013), *generative models* (GM) (Kingma & Welling, 2013), and *hybrid methods* (HB) (Sohn et al., 2020), within the C2ST framework. However, the result was not satisfactory (see Table 1), because two-sample testing fundamentally differs from typical classification tasks; it is a problem of distinguishing between two distributions (or saying two samples) rather than an instance-level classification. Furthermore, *the high degree of overlap between the two samples* in testing dataset challenges the basic assumptions of these SSL methods, such as HDGM in Figure 2b and Figure 2c, where two distributions are largely overlapped, violating the assumptions of many SSL methods.

This violation of assumptions leads SSL methods to have low test power in two-sample testing. Our method, SSL-C2ST, is implemented in two phases. At first, we learn the IRs from whole dataset in an unsupervised autoencoder-based representation learning (Tschannen et al., 2018a). Following this, we apply the C2ST framework, not by training an randomly initialized classifier, but by fine-tuning the pre-trained encoder with an added classification layer in order to learn the DRs.

We provide the first theoretical analysis to show that, with a high probability, involving a larger testing set (without sample information) in the training process will lead to a higher lower bound of the test power of SSL-C2ST, verifying the effectiveness of SSL-C2ST in theory. Besides, the empirical test power on three benchmark datasets also shows that SSL-C2ST clearly outperforms C2ST.

On the empirical side, we are also interested in the data representations extracted by the trained classifier in SSL-C2ST. We perform MMD tests (with a linear kernel) on the different-level data representations of the testing set, called SSL-C2ST-M. These tests clearly outperforms the corresponding baselines empirically. Notably, SSL-C2ST-M outperforms the *MMD with the deep kernel* (MMD-D (Liu et al., 2020)) and MMD-FUSE (Biggs et al., 2023) on the MNIST and ImageNet dataset.

**Impact of our study in the field.** The success of SSL-C2ST(-M) might provide evidence that SSL-based testing methods have the potential to overcome the key issues of both paradigms. For the supervised paradigm, SSL-based testing methods can leverage the useful information in the testing set (without sample information), thus we can expect either to use a smaller training set (sacrificing less power) or to learn a better function/kernel (more extra power) with the help of the useful information in the testing set. For the unsupervised paradigm, SSL-based testing methods might provide some supervision to guide the learning process of unsupervised learning or to form a better candidate set that contains a function/kernel that can help distinguish between two samples better.

Table 1: Result of C2ST test power on HDGM-Easy, HDGM-Medium and HDGM-Hard (d=10), on different total size of two samples $N$ inputed in 100 trials. Compared to other application of SOTA SSL methods on C2ST, where C2ST-CR, C2ST-PL, C2ST-GM, and C2ST-HB represent that we learn the classifier of C2ST using consistency-regularisation, pseudo-labelling, generative-model, and hybrid SSL frameworks, respectively, and SSL-C2ST is our method. [2]

| Method | HDGM-Easy | | | HDGM-Medium | | | HDGM-Hard | | |
|--------|-----------|------|-------|-------------|---------|---------|-----------|---------|---------|
| | N=60 | N=80 | N=100 | N=2000 | N=3000 | N=4000 | N=4000 | N=6000 | N=8000 |
| C2ST | 0.64 | 0.91 | 0.99 | 0.44 | 0.82 | 0.97 | 0.29 | 0.49 | 0.78 |
| C2ST-CR | 0.65 | 0.92 | **1.00** | 0.40 | 0.84 | 0.97 | 0.32 | 0.42 | 0.75 |
| C2ST-PL | 0.72 | 0.96 | 0.99 | 0.40 | 0.76 | 0.93 | 0.36 | 0.45 | 0.77 |
| C2ST-GM | 0.64 | 0.92 | **1.00** | 0.43 | 0.85 | 0.97 | 0.22 | 0.40 | 0.72 |
| C2ST-HB | **0.99** | **1.00** | **1.00** | 0.25 | 0.43 | 0.58 | 0.28 | 0.43 | 0.65 |
| SSL-C2ST | 0.97 | 0.99 | **1.00** | **0.58** | **0.97** | **1.00** | **0.50** | **0.81** | **0.99** |

## 2 PRELIMINARY

**Two-sample Test.** Two-sample test is one of the statistical hypothesis tests that aims to assess whether two independent identically distributed $i.i.d.$ samples, denoted by $S_P = \{x_i\}_{i=1}^n \sim \mathbb{P}^n$ and $S_Q = \{y_j\}_{j=1}^m \sim \mathbb{Q}^m$, where $x_i, y_j \in \mathcal{X}$, are drawn from the same distribution (Lehmann & Romano, 2005). In two-sample testing, the *null hypothesis* $H_0$ refers to two samples sourcing from the same distribution, which stands for $\mathbb{P} = \mathbb{Q}$. The *alternative hypothesis* $H_1$ indicates that two samples are drawn from different distributions, meaning $\mathbb{P} \neq \mathbb{Q}$. Whether we should accept or reject $H_0$ depends on the test statistic $\hat{t}$, which represents the differences between two samples.

**Classifier Two-sample Test (C2ST).** The idea of C2ST is to use a supervised classification algorithm to distinguish between the two samples. If the classifier performs significantly better than random guessing, it suggests that two samples come from different distributions (Lopez-Paz & Oquab, 2018b):

**Problem Setting.** In our problem setting, we assume the total number of two samples are *fixed and given*, and we are trying to distinguish whether these two given samples are from the same distribution or not. No more extra data is provided for testing data and the test data is known, so it can be regarded as a *transductive learning problem*. Thus, the C2ST is conducted in the following steps:

Firstly, construct the dataset $\mathcal{S} = \{(x_i, 0)|x_i \in S_P\}_{i=1}^n \cup \{(y_j, 1) \in S_Q\}_{j=1}^m := \{(z_k, l_k)\}_{k=1}^{m+n} \sim \mathcal{D}$, where $m = n$; then shuffle and split $\mathcal{S}$ into $\mathcal{S}_{\text{tr}}$ and $\mathcal{S}_{\text{te}}$, where $\mathcal{S} = \mathcal{S}_{\text{tr}} \cup \mathcal{S}_{\text{te}}$. Let $f^* : \mathcal{X} \to \{0, 1\}$ be a binary classifier that is trained on $\mathcal{S}_{\text{tr}}$ from a concept class $\mathcal{C}$ and $p_k = p(l_k = 1|z_k)$, where

$$f^* = \arg\min_{f \in \mathcal{C}} \sum_{(z_k, l_k) \in \mathcal{S}} -\left[l_k \log p_k + (1 - l_k) \log (1 - p_k)\right],$$

and $f^*(z_k)$ be the estimate of the conditional probability distribution $\mathbb{I}\left(p(l_k = 1|z_k) > \frac{1}{2}\right)$, the statistic or the accuracy of the classifier $f^*$ on $\mathcal{S}_{\text{te}}$ can be written as:

$$\hat{t} = \frac{1}{n_{\text{te}}} \sum_{(z_k, l_k) \in \mathcal{S}_{\text{te}}} \mathbb{I}\left[f^*(z_k) = l_k\right], \tag{1}$$

where $n_{\text{te}} = |\mathcal{S}_{\text{te}}|$ and $\mathbb{I}$ is the indicator function. Finally, we compute the $p$-value to determine if the test statistic is significantly greater than the random guessing accuracy, utilizing the approximate null distribution of C2ST outlined in Appendix D.1 and the permutation test discussed next.

**Testing with $\hat{t}$.** According to the standard central limit theorem (Serfling, 2009), the test statistic $\hat{t}$ in Eq. (1) converges to normal distributions under both the null or alternative hypothesis. Although it is viable for us to derive the threshold $t_\alpha$ of the null hypothesis distribution and perform a traditional Z-Test, it is simpler and faster to instead implement a permutation test (Sutherland et al., 2017). We will permute and randomly assign samples to new $S_P^{\text{te}'}$ and $S_Q^{\text{te}'}$ for $n$ times. Under $H_0$, the samples from $\mathbb{P}$ and $\mathbb{Q}$ should be interchangeable, implying that the test statistic $\hat{t}$ should exhibit minimal variation between its value based on the original sequence of samples and its computation from several randomly permuted sequences. Thus, if the original test statistic is large enough than most of the statistic derived from the randomly permuted sequences, we can conclude that we reject $H_0$.

**C2ST-based MMD (C2ST-M).** Moreover, we can also consider using a trained classifier in C2ST to extract representations of two samples, and then regard representations of two samples as the new two samples. For these new two samples, we can use MMD (with a linear kernel) to compute the difference between two samples. Let $S_P^{\text{te}}$ and $S_Q^{\text{te}}$ be the splitting samples of $S_P$ and $S_Q$ in the testing set $\mathcal{S}_{\text{te}}$ and $n_x^{\text{te}}$ and $n_y^{\text{te}}$ be the sample size of $S_P^{\text{te}}$ and $S_Q^{\text{te}}$. In general, the statistic used in C2ST-M is

$$\hat{t}_M = \left\| \frac{1}{n_x^{\text{te}}} \sum_{x_i \in S_P^{\text{te}}} h(x_i) - \frac{1}{n_y^{\text{te}}} \sum_{y_i \in S_Q^{\text{te}}} h(y_i) \right\|_2^2, \tag{2}$$

where $h$ is the feature extractor (could be the model's output, i.e., logit), or the model's hidden-layer output, and $\|\cdot\|_2$ is the L2 norm. When $h$ is logits, C2ST-M is known as C2ST-L in (Liu et al., 2020).

## 3 REVISIT NON-PARAMETRIC TWO-SAMPLE TEST AS A SEMI-SUPERVISED LEARNING PROBLEM

This section presents two research questions that we will address in the paper. As both existing two-sample testing paradigms have their own limitations, our first research question comes out

### 3.1 IS IT POSSIBLE TO BOTH ELIMINATE THE SIDE-EFFECT OF DATA SPLITTING AND OBTAIN THE HIGH DISCRIMINATIVE POWER?

Except the supervised paradigm and unsupervised paradigm, the semi-supervised one is another well-known paradigm. According to the definition of SSL, *SSL can leverage the information $P(x)$ from unlabeled data to help the inference of $P(y|x)$* (Chapelle et al., 2006). If the unlabeled data degrades prediction accuracy by misguiding the inference (e.g., due to violating the assumptions of SSL techniques), then that cannot be classified as effective SSL method. As we attempt to utilize the information from the unlabeled testing data to increase the test power of the supervised two-sample testing methods, SSL techniques seem to be reliable to solve that research question. However, since we are the first to frame two-sample testing as a SSL problem, we have to be responsible to evaluate whether current SSL techniques can be *directly* applied on the supervised two-sample testing methods.

### 3.2 CAN SOTA SSL TECHNIQUES BE SUCCESSFULLY APPLIED ON SUPERVISED TWO-SAMPLE TESTING METHODS?

This question is worthy to investigate, since in the definition of SSL, the consequence of failure in applying SSL techniques is highlighted, which can lead to a worse performance than the original supervised method. Thus, we will firstly conduct motivation experiments to *directly* apply the main SOTA SSL techniques on the SOTA supervised two-sample testing method C2ST to examine the fitness of SSL assumptions on the two-sample testing data. If it fails, we will propose a viable method that can utilize the information from the unlabeled testing dada, which can pave the way for the further advanced techniques to be applied.

## 4 CAN WE DIRECTLY APPLY SSL METHODS IN TWO-SAMPLE TESTING?

In this section, we will discuss the key assumptions of traditional SSL methods. Then, we will analyze whether we can directly apply those methods in our two-sample testing scenarios.

**Assumptions of SSL methods.** In principle, incorporating unsupervised information from unlabeled data has the potential to enhance the predictions made by purely supervised learning models. However, the efficacy of SSL is often relied on some assumptions below (Chapelle et al., 2006).

- *Smoothness assumption:* If points $x_1$ and $x_2$ are close, then so should be their labels $y_1$, $y_2$.
- *Cluster assumption:* If points are in the same cluster, they are likely to be of the same class.
- *Manifold assumption:* The (high-dimensional) data lie (roughly) on a low-dimensional manifold.

---

[2]The result does not include standard deviation, since each trial we are testing whether two groups of drawn sample are from same distribution, and the result of each trial is either 0 or 1.

Based on those assumptions, there are five representative SSL frameworks (Yang et al., 2023): consistency-regularisation (Xie et al., 2020), pseudo-labelling (Lee et al., 2013), graph-based (Song et al., 2021b), generative-models (Kingma & Welling, 2013) and hybrid (Sohn et al., 2020) SSL methods. The details of SSL methods are demonstrated in Appendix B.

**Testing data might not satisfy the assumptions made by many SSL methods.** In the traditional two-sample testing problem settings, there are normally overlapping between two samples. As we can see in Figure 2b and Figure 2c, for the HDGM-Medium and HDGM-Hard datasets, there are high-overlapping areas between two distributions. This will *highly violate the first two assumptions* of SSL mentioned above. For the smoothness assumption, our dataset will have exactly the same data point in two samples, but allocated with different labels, this will notably influence the SSL methods that based on such assumption. For cluster assumptions, we can see in HDGM-Medium, although there are two obvious clusters, they are not have same labels within the same cluster.

**Empirical result for validity of SOTA SSL methods on two-sample testing.** The empirical results, presented in Table 1[3], show that the application of SOTA SSL methods on C2ST not only underperforms our proposed method but also often yields poorer results compared to the original C2ST on HDGM-Medium and HDGM-Hard datasets, which are the common overlapping distribution data in the context of two-sample testing. This underperformance can be attributed to the fundamental nature of the two-sample testing problem, which is distinct from typical classification tasks. In two-sample testing, the two input samples should not inherently possess class labels. During training, we manually assign labels to facilitate distinction by the classifier, whereas in testing, we consider the two samples holistically rather than focusing on individual instance accuracy. Furthermore, standard SSL methods, which primarily enhance classification through data augmentation based on smoothness assumptions or infer pseudo labels based on clustering assumptions, aim to generate high-confidence training data. However, in two-sample testing, these approaches are flawed; data augmentation may alter the samples' distributions, and pseudo label inference often proves inaccurate. These discrepancies lead to the ineffectiveness of these SSL methods in two-sample testing contexts. Therefore, we propose a two-sample test through a two-phase SSL approach, shown below.

## 5  HOW TO UTILIZE UNLABELLED DATA INCREASING TEST POWER?

In this section, we introduce the structure design and the algorithm of our SSL-C2ST, and then we offer theoretical analysis to validate the effectiveness of our method.

### 5.1  OUR PROPOSAL: SSL-C2ST

As the two-sample testing problem violates the native assumptions of SOTA SSL methods, we propose a pipeline that follows the definition of SSL, which utilizes the unlabelled samples and labelled samples in two phases. The first phase is an unsupervised auto-encoder-based (AE-based) representation learning, which learns a feature extractor that captures the inherent features for both samples. The next phase is the same as the C2ST pipeline, where the feature encoder in the model is not randomly initialized, but extracted from the representation learning in the previous phase. The ablated part of this approach compared to the C2ST is the AE-based representation learning, so its effectiveness will be aligned with AE-based representation learning, which relies on the manifold assumption, and that is particularly well-suited for two-sample testing scenarios The paradigm of SSL-C2ST is shown in Figure 3, consisting of three steps: 1) learning IRs; 2) learning DRs; and 3) performing two-sample testing.

Since SSL-C2ST has the *same* classifier architecture as C2ST but with *different* training objectives, we need to decompose the classifier model $f$ into two parts: a feature extractor $\phi \in \mathcal{F} : \mathcal{X} \to \mathbb{R}^k$ that used to learn IRs and followed by a classifier $g \in \mathcal{G} : \mathbb{R}^k \to \{0, 1\}$ that used to learn DRs. We denote by $\phi_f$ and $g_f$ the feature extractor and the classifier of a specified model $f$. Then, let $f' \in \mathcal{C}_\phi : \mathcal{X} \to \{0, 1\}$ be the SSL-C2ST classifier model, where $\mathcal{C}_\phi = \{f'|f' = g \circ \phi, g \in \mathcal{G}\} \subseteq \mathcal{C}$ and $\mathcal{C} = \bigcup_{\phi \in \mathcal{F}} \mathcal{C}_\phi$. Given two available samples $S_P$ and $S_Q$, and construct a dataset $\mathcal{S}$ referred to the problem setting in Section 2.

---

[3]The experimental details of this table can be found in Appendix B, where all SSL methods and how to use these methods in testing are introduced.

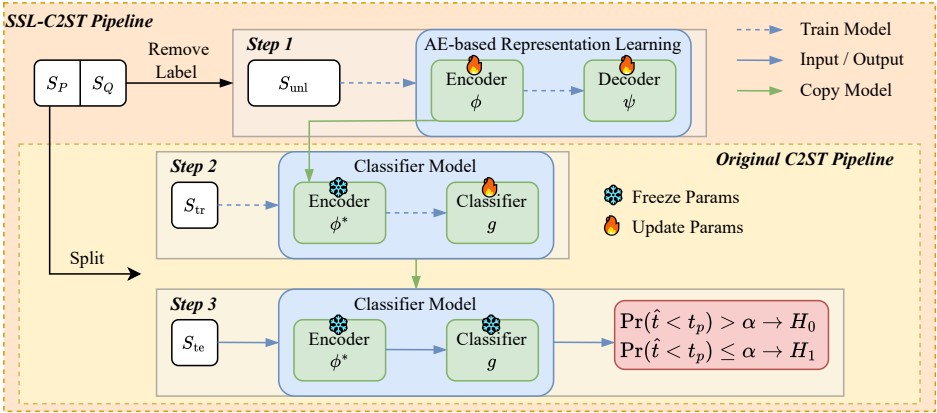

Figure 3: Overview of the SSL-C2ST paradigm compared to original C2ST paradigm. Firstly, an encoder was learned from unsupervised AutoEncoder-Based representation learning on whole data. Secondly, fine-tune the learned encoder followed by the supervised learning in C2ST. At last, perform the permutation test based on the model and the output statistics.

**Learning IRs.** The first step is to train a representation learning model on the whole unlabelled dataset $\mathcal{S}_{\text{unl}}$, using the *mean squared error* (MSE) as the loss function to compare the differences between input and reconstructed output. Specifically, we aim to learn a function $\phi^*$ such that

$$\phi^* = \arg\min_{\phi} \ \frac{1}{|\mathcal{S}_{\text{unl}}|} \sum_{z_i \sim \mathcal{S}_{\text{unl}}} \|\psi(\phi(z_i)) - z_i\|_2^2, \tag{3}$$

where $\psi : \mathbb{R}^k \to \mathcal{X}$ is the decoder, and $\phi^*(z_i)$ is called the IR of $z_i$.

**Learning DRs.** Then, utilize the featurizer from the representation learning model and concatenate with a classification layer to form a binary classifier model. The combined model is fine-tuned on $\mathcal{S}_{\text{tr}}$, applying a *cross-entropy* (CE) loss function to compare the output of SSL-C2ST with the label of samples. Specifically, we aim to learn a function $g^*$ such that $\Pr_{(z_i,l_i)\sim\mathcal{S}} [g \circ \phi^*(z_i) \neq l_i]$ can be minimized on $S$, which can be implemented by the following surrogate objective.

$$g^* = \arg\min_{g} \mathcal{L}(g \circ \phi^*) = \arg\min_{g} \frac{1}{|\mathcal{S}_{\text{tr}}|} \sum_{(z_i,l_i)\sim\mathcal{S}_{\text{tr}}} -[l_i \log p_i + (1 - l_i)(1 - \log p_i)], \tag{4}$$

where $p_i$ is defined in the problem setting in Section 2. In this paper, $g^*$ is a neural network consisting of multiple layers, so $g^*$ can be expressed by $g^* = h^* \circ h^*_{\text{rep}}$ where $h^*_{\text{rep}} \in \{h_{\text{rep}} : \mathbb{R}^k \to \mathbb{R}^{d_{\text{rep}}}\}$ and $h^* \in \{h : \mathbb{R}^{d_{\text{rep}}} \to \{0,1\}\}$. Normally, we call $h^*$ as a classification head and $h^*_{\text{rep}}$ as a representation function. Thus, a DR of $z_i$ is $h^*_{\text{rep}} \circ \phi^*(z_i)$.

**Testing.** In the end, compute the test statistic in Eq. (1) (by setting $f'^*$ as $g^* \circ \phi^*$) based on the original sequence of samples and the $r$ times permuted samples, reject $H_0$ if original statistic is larger than the threshold derived from permuted statistics.

**Overall algorithm.** The procedure of how to implement SSL-C2ST is summarised in Algorithm 1 of the Appendix A. Note that, in Eq. (4), we can also consider continuing optimizing $\phi^*$, which can increase the complexity of trainable classifiers, if the data is very complex.

## 5.2 THEORETICAL ANALYSIS OF SSL-C2ST

In this section, we discuss what the approximated power of our SSL-C2ST test is and how the size of unlabelled data $m_{\text{u}}$ helps to improve test power.

**Test Power.** Test power is the probability that a test will correctly reject $H_0$, when $H_1$ holds. It represents the ability of the test to detect the difference between $\mathbb{P}$ and $\mathbb{Q}$, so analyzing this power is essential for evaluating the performance of one two-sample testing method.

**Theorem 5.1.** *(Lopez-Paz & Oquab, 2018b) Let $f' \in \mathcal{C}_\phi : \mathcal{X} \to \{0, 1\}$ be the SSL-C2ST classifier model. Let $H_0 : t = \frac{1}{2}$ and $H_1 : t = 1 - \epsilon(\mathbb{P}, \mathbb{Q}; f')$, where $t$ is the test accuracy and $\epsilon(\mathbb{P}, \mathbb{Q}; f') = \Pr_{(z_i, l_i) \sim \mathcal{D}} [f'(z_i) \neq l_i] / 2 \in \left(0, \frac{1}{2}\right)$ represents the inability of $f'$ to distinguish between $\mathbb{P}$ and $\mathbb{Q}$. The test power of $\hat{t}$ is:*

$$\Pr_{H_1} \left( \hat{t}_{H_0} > t_\alpha \right) = \Phi \left( \frac{\left(\frac{1}{2} - \epsilon(\mathbb{P}, \mathbb{Q}; f')\right) \sqrt{n_{\text{te}}} - \Phi^{-1}(1 - \alpha)/2}{\sqrt{\epsilon(\mathbb{P}, \mathbb{Q}; f') - \epsilon(\mathbb{P}, \mathbb{Q}; f')^2}} \right), \tag{5}$$

*where $\alpha \in (0, 1)$ is the significance level, $t_\alpha$ is the $(1 - \alpha)$ quantile and $\Phi$ is the CDF of standard normal distribution. The Type-I error of $\hat{t}$ is also controlled no more than $\alpha$, which ensures that the test will not always reject $H_0$, when $H_0$ is true.*

**Understand SSL-C2ST via Theorem 5.1.** In hypothesis testing, our primary aim is to maximize test power while maintaining control over the Type-I error rate. While we know that via Theorem 5.1, $\Phi^{-1}(1 - \alpha)/2$ is a constant, for a reasonably fixed large $n_{\text{te}}$, the first term $(\frac{1}{2} - \epsilon(\mathbb{P}, \mathbb{Q}; f'))$ in the numerator dominates the test power. In fact, to ensure that the model can achieve the optimal test power on a fixed test dataset, it is equivalent to minimize

$$\mathcal{J}(\mathbb{P}, \mathbb{Q}; f') := \epsilon(\mathbb{P}, \mathbb{Q}; f') \Big/ (1 - \epsilon(\mathbb{P}, \mathbb{Q}; f')), \tag{6}$$

where we estimate it with

$$\hat{\mathcal{J}}(S_P, S_Q; f') := \hat{\epsilon}(S_P, S_Q; f') \Big/ (1 - \hat{\epsilon}(S_P, S_Q; f')), \text{ and } \hat{\epsilon}(S_P, S_Q; f') \in \left(0, \frac{1}{2}\right), \tag{7}$$

where $\hat{\epsilon}(S_P, S_Q; f') = \frac{1}{2}\widehat{err}(f') = \frac{1}{2|\mathcal{S}|} \sum_{(x_i, l_i) \sim \mathcal{S}} \mathbb{I}[f'(x_i) \neq l_i]$. The proof of above can be found in Appendix D.1. From Eq. (7), we can find that if we learn a classifier $f'$ from Eq. (4) that has a smaller $\hat{\epsilon}(S_P, S_Q; f')$, we can minimize the $\hat{\mathcal{J}}$, leading to maxmizing the test power. Thus, we will analyze how the use of unlabelled data and the size of unlabelled data $m_{\text{u}}$ helps to learn a classifier model $f'$ that have a smaller $\hat{\epsilon}(S_P, S_Q; f')$ in the semi-supervised learning.

We first give a definition of compatibility, an important measurement when analyzing SSL methods.

**Definition 5.2** (Compatibility). *The compatibility of classifier model $f$ is defined as $\chi : \mathcal{C} \times \mathcal{X} \to [0, 1]$, and $\chi(f, \mathcal{S}) = \mathbb{E}_{x \sim \mathcal{S}}[\chi(f, x)]$ estimates how "compatible" the feature extractor of $f$ is with fixed dataset $\mathcal{S}$. Thus, the incompatibility of $f$ with $\mathcal{S}$ is $1 - \chi(f, \mathcal{S})$. We can also call it unlabelled error rate $err_{\text{unl}}(\phi_f)$, where $err_{\text{unl}}(\phi_f) = 1 - \chi(f, \mathcal{S})$. Thus, given value $\xi$, we define $\mathcal{C}_{\mathcal{S}, \chi}(\xi) = \{f \in \mathcal{C} : err_{\text{unl}}(\phi_f) \leq \xi\}$.*

Then, the following theorem shows our main theoretical result, based on the compatibility.

**Theorem 5.3.** *Let $f'^* = \arg\min_{f' \in \mathcal{C}_\phi} [\epsilon(\mathbb{P}, \mathbb{Q}; f') | err_{\text{unl}}(\phi_{f'}) \leq \xi]$. The following holds with probability at least $1 - \delta$, for any arbitrarily small $\Delta_{m_{\text{u}}, m_{\text{l}}} > 0$,*

$$\hat{\epsilon}(S_P, S_Q; f') \leq \epsilon(\mathbb{P}, \mathbb{Q}; f'^*) + \frac{\Delta_{m_{\text{u}}, m_{\text{l}}}}{2} + \sqrt{\frac{\ln\left(\frac{4}{\delta}\right)}{8m_{\text{u}}}}, \tag{8}$$

*with the unlabelled sample size*

$$m_{\text{u}} = \mathcal{O}\left(\Delta^{-2} \log \Delta^{-1} \max\left[VCdim(\mathcal{C}_\phi), VCdim(\chi(\mathcal{C}_\phi))\right] + \Delta^{-2} \log(2/\delta)\right),$$

*and the labelled sample size*

$$m_{\text{l}} = \frac{8}{\Delta^2}\left[\log\left(2\mathcal{C}_{\phi, \mathcal{S}, \chi}(\xi + 2\Delta)[2m_{\text{l}}, \mathcal{S}]\right) + \log(4/\delta)\right].$$

*Here, $\chi(\mathcal{C}_\phi) = \{\chi_{f'} : f' \in \mathcal{C}_\phi\}$ is assumed to have a finite VC dimension, $\chi_{f'}(\cdot) = \chi(f', \cdot)$, and $\mathcal{C}_{\phi, \mathcal{S}, \chi}(\xi + 2\Delta)[2m_{\text{l}}, \mathcal{S}]$ is the expected split number for $2m_{\text{l}}$ points drawn from $\mathcal{S}$ using functions in $\mathcal{C}_\phi \cap \mathcal{C}_{\mathcal{S}, \chi}(\xi + 2\Delta)$.*

The proof of Theorem 5.3 is presented in the Appendix D.2. Theorem 5.3 indicates that when we are training model $f'$, the increment in the size of unlabelled data $m_{\text{u}}$ can reduce the upper bound

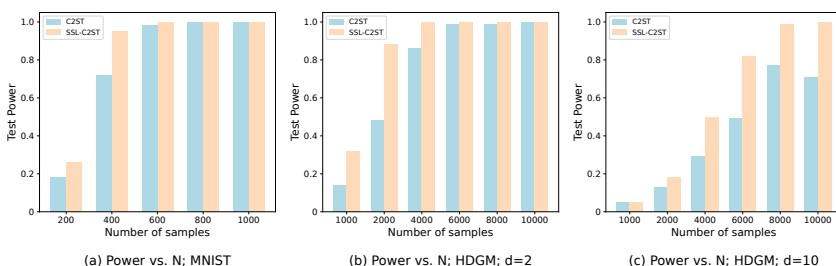

(a) Power vs. N; MNIST

(b) Power vs. N; HDGM; d=2

(c) Power vs. N; HDGM; d=10

Figure 4: Test power of SSL-C2ST and C2ST. Barplot to show how SSL-C2ST outperforms C2ST in the *MNIST* dataset (a), *HDGM-D* when $d = 2$ (b) and *HDGM-D* when $d = 10$ (c).

of the empirical error rate of $f'$. Thus, the upper bound of the empirical inability $\hat{\epsilon}(S_P, S_Q; f')$ will decrease as well, leading to a direct increase on the lower bound of the approximate test power as shown in Eq. (7). In other words, this theorem ensures the effectiveness of unlabelled data in the improvement of SSL-C2ST test power.

## 6 EXPERIMENTS

In this section, we will analyze the experiment result of SSL-C2ST on two commonly used benchmark datasets and one advanced ImageNet dataset. Also, we will discuss the empirical extensions based on SSL-C2ST (i.e., SSL-C2ST-M), and present the comparative analysis of experimental results across three benchmarks. The overview information of three datasets and experimental implementation details of proposed method and other two-sample testing methods can be found in Appendix C.

**Datasets.** We conducted experiments on five different datasets to thoroughly evaluate our methods. To assess the performance of current SSL methods applied to two-sample testing, we utilized three synthetic datasets: *HDGM-Easy*, *HDGM-Medium*, and *HDGM-Hard*. Moreover, we conduct the experiments of our proposed methods against other SOTA two-sample testing methods to evaluate the effectiveness of semi-supervised paradigm on three datasets: *MNIST*, *ImageNet*, and *HDGM*. Detailed descriptions of these datasets are provided in Appendix C.1.

**Baselines.** We evaluate the performance of our proposed methods SSL-C2ST and SSL-C2ST-M against several SOTA baseline methods in two-sample testing, specifically C2ST, C2ST-M, MMD-D, and MMD-FUSE. These baselines serve as competitive references to highlight the improvements achieved by our approach. For comprehensive details on each baseline method, including their implementations and parameter settings, please refer to Appendix C.2 and C.3.

**Ablation Study: Verification of SSL-C2ST over C2ST**. We first verify the effectiveness of SSL-C2ST via comparing SSL-C2ST and C2ST, which provides empirical evidence for Theorem 5.3. The implementation details of SSL-C2ST and C2ST can be found in Appendix C.2.

The visualized result of how our SSL-C2ST outperforms C2ST is displayed in Figure 4. In both dataset *MNIST* and *HDGM-Hard*, we can see that the test power of SSL-C2ST is higher than that of C2ST no matter how many numbers of two samples are drawn from the distribution. Although the differences between two methods are little when $N$ is small, the test power of SSL-C2ST has a huge gap over C2ST when $N$ is large enough and converges to 1 with a relative smaller $N$ compare to C2ST. This also verifies our theoretical analysis that our model is more likely to have large improvement of test power if the number of unlabelled samples in the semi-supervised learning is sufficiently large.

Compared to C2ST, SSL-C2ST learns a compact and potentially more informative representation of the whole data, which makes efficient use of the unlabelled test data. This can not only discover underlying patterns or features that might not directly related to the labels but to the data distribution itself, but also provide a regularizing effect to prevent the model being more likely to overfit the training data. Such featurizer in the SSL-C2ST can result in a better generalization from the learned representations and improve the classifier's performance on the testing set predictions.

**Test-power Results of SSL-C2ST-M**. After we validate the effectiveness of SSL-C2ST, we will also introduce an advanced empirical testing method based on our SSL-C2ST: SSL-C2ST-M and how they

---

[3]The result does not include standard deviation, since each trial we are testing whether two groups of drawn sample are from same distribution, and the result of each trial is either 0 or 1.

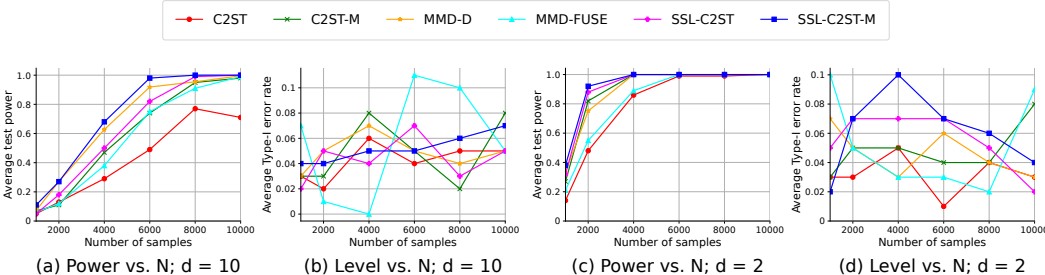

(a) Power vs. N; d = 10  (b) Level vs. N; d = 10  (c) Power vs. N; d = 2  (d) Level vs. N; d = 2

Figure 5: Results on *HDGM-D* and *HDGM-S* for $\alpha = 0.05$. Left: average test power (a) and average type-I error (b) when increasing total two sample size $N$ from $N = 1000$ to $N = 10000$, keeping $d = 10$ in 100 trials. Right: average test power (c) and average type-I error (d) when increasing $N$ from $N = 1000$ to $N = 10000$, keeping $d = 2$ in 100 trials.

Table 2: MNIST and ImageNet ($\alpha = 0.05$). Average test power for comparing $M$ real MNIST images to $M$ DCGAN-generated MNIST images, and Average test power for comparing $M$ real ImageNet images to $M$ StyleGAN-XL-generated ImageNet images. [5]

| Method | MNIST | | | | | | ImageNet | | | | | |
|---|---|---|---|---|---|---|---|---|---|---|---|---|
| | M=200 | M=400 | M=600 | M=800 | M=1000 | Avg. | M=200 | M=400 | M=600 | M=800 | M=1000 | Avg. |
| C2ST | 0.180 | 0.720 | 0.980 | **1.000** | **1.000** | 0.776 | 0.150 | 0.300 | 0.350 | 0.600 | 0.850 | 0.450 |
| C2ST-M | 0.250 | 0.730 | 0.990 | **1.000** | **1.000** | 0.794 | 0.150 | 0.350 | 0.450 | 0.700 | 0.850 | 0.500 |
| MMD-D | 0.290 | **0.996** | **1.000** | **1.000** | **1.000** | 0.857 | 0.210 | 0.400 | 0.570 | 0.780 | **1.000** | 0.592 |
| MMD-FUSE | 0.320 | 0.870 | **1.000** | **1.000** | **1.000** | 0.838 | 0.230 | 0.450 | 0.610 | **0.790** | **1.000** | 0.616 |
| SSL-C2ST | 0.260 | 0.950 | **1.000** | **1.000** | **1.000** | 0.842 | 0.200 | 0.400 | 0.500 | 0.650 | 0.950 | 0.540 |
| SSL-C2ST-M | **0.491** | 0.985 | **1.000** | **1.000** | **1.000** | **0.895** | **0.400** | **0.500** | **0.650** | 0.750 | **1.000** | **0.660** |

outperform state-of-the-art testing methods from the supervised paradigm (MMD with deep kernel) and the unsupervised paradigm (MMD-FUSE). The implementation details of these MMD-based methods can be found in Appendix C.3.

The overall result of all testing methods for the HDGM dataset is shown in Figure 5. We can see that SSL-C2ST-M method has the highest test power in both the 2-dimensional *HDGM-D* and 10-dimensional *HDGM-D*, no matter how we choose $N$, while all type-I errors are reasonably controlled around $\alpha = 0.05$. For *MNIST* and *ImageNet* datasets, the results of all methods are shown in Table 2, although SSL-C2ST-M does not outperform MMD-D and MMD-FUSE in *MNIST* when $N = 400$ and in *ImageNet* when $N = 600$, it has a clear increase in the test power when $N$ is small, leading to a powerful average test power performance across two image datasets.

**Discussion of Sequential Two-sample Testing.** Moreover, sequential two-sample testing methods also utilize information from the test data but is a different problem setting from ours. We provide detailed descriptions of sequential two-sample testing in Appendix C.6, along with experimental results C.7 demonstrating that our methods outperform these approaches within our setting. Additionally, we discuss how our proposed paradigm can be applied to other supervised two-sample testing methods, potentially enhancing their performance in two-sample testing.

## 7  CONCLUSION

Non-parametric two-sample testing is an important problem in both statistics and machine learning fields. This paper presents a new paradigm, *semi-supervised learning-based classifier two-sample test* (SSL-C2ST), to learn better data representations for addressing this problem and gives a theoretical analysis of why the proposed paradigm can have a higher test power compared to two representative paradigms in the field. In the end, an advanced empirical testing method, *SSL-C2ST with MMD* (SSL-C2ST-M), is presented in the experiments and shows superior performance compared to previous testing methods. Both theoretical analysis and empirical evidence show that the proposed new paradigm might be a cure for key issues of the existing two paradigms in the two-sample testing field, paving a new road to revisit and address the two-sample testing problem.

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

## A ALGORITHM

We present the complete algorithm for SSL-C2ST in Algorithm 1.

---

**Algorithm 1** paradigm of testing with SSL-C2ST

---

**Input:** $S_P$, $S_Q$, $\mathcal{S}$, significance level $\alpha$, latent feature vector size $H$, an autoencoder $f_a$ consist of a featurizer $\phi$ and a decoder $\phi^{-1}$ parameterized by $\theta_\phi$, a binary classifier $g$ concatenated after featurizer parameterized by $\theta_g$, SSL-C2ST model $f = g \circ \phi$, learning rate $\eta_\phi, \eta_g$, MSE loss function $\mathcal{L}_{\text{MSE}}$, CE loss function $\mathcal{L}_{\text{CE}}$, total epoch for representation learning $T_{\text{rl}}$, total epoch for training classifier $T_{\text{cl}}$.

**1:** Derive the unlabelled data $\mathcal{S}_{\text{unl}} = \text{shuffle}(S_P \cup S_Q)$

*# Phase 1: train the Featurizer $\phi$ from $f_a$ on $\mathcal{S}_{\text{unl}}$*

**for** $t = 1, 2, \ldots, T_{\text{rl}}$ **do**

    **2:** $X_t \leftarrow$ minibatch from $\mathcal{S}_{\text{unl}}$;

    **3:** $X'_t \leftarrow f_a(X_t)$;

    **4:** $\theta_\phi \leftarrow \theta_\phi - \eta_\phi \nabla_{\text{Adam}} \mathcal{L}_{\text{MSE}}(X_t, X'_t)$ based on Eq. (3);

**end for**

*# Phase 2: train a Classifier $f$ (consist of $\phi$ and $g$) on $S^{\text{tr}} = (S_P^{\text{tr}}, \mathbf{0}) \cup (S_Q^{\text{tr}}, \mathbf{1})$*

**for** $t = 1, 2, \ldots, T_{\text{cl}}$ **do**

    **5:** $(X_t, l_t) \leftarrow$ minibatch from $\mathcal{S}^{\text{tr}}$;

    **6:** $\hat{l}_t \leftarrow g \circ \phi(X_t)$;

    **7:** $\theta_g \leftarrow \theta_g - \eta_g \nabla_{\text{Adam}} \mathcal{L}_{\text{CE}}(\hat{l}_t, l_t)$ based on Eq. (4);

**end for**

*# Phase 3: permutation test with $f$ on $S^{\text{te}} = S_P^{\text{te}} \cup S_Q^{\text{te}}$*

**8:** $est \leftarrow \hat{t}(S_P^{\text{te}}, S_Q^{\text{te}}; f)$ based on Eq. (1);

**for** $i = 1, 2, \ldots, n_{\text{perm}}$ **do**

    **9:** Shuffle $S^{\text{te}}$ into $X$ and $Y$;

    **10:** $perm_i \leftarrow \hat{t}_{\text{M}}(X, Y; f)$

**end for**

**Output:** $\mathbb{I}\left[ \frac{1}{n_{perm}} \sum_{i=1}^{n_{perm}} \mathbb{I}(est < perm_i) \leq \alpha \right]$

---

## B OVERVIEW OF MAJOR CATEGORIES OF SOTA SSL METHODS

Building on the SSL assumptions, we will recap how contemporary SOTA SSL methods incorporate these principles and assumptions, setting the stage for an analysis of their applicability to the specific challenges presented by our problem setting.

**Transductive vs Inductive learning.** Classification tasks within machine learning can typically be categorized within two distinct problem settings: transductive and inductive learning (Chapelle et al., 2006). *Transductive learning* is concerned with predicting the labels of the specific unlabeled data that was present during the training process, emphasizing a tailored fit to this data. *Inductive learning*, on the other hand, focuses on the generalization of the learned classifier to new, unseen data. In learnable two-sample testing, the goal is to test whether the given two samples are drawn from same distributions. To make it, we firstly split samples into labelled set and unlabelled set, then find out that whether it is possible to learn a classifier that can distinguish two samples from the mixed unlabelled set. It becomes apparent that applying SSL methodologies to the two-sample testing problem inherently requires a transductive learning approach. This conceptual groundwork necessitates a detailed examination of current SSL methods to identify their foundational assumptions and evaluate their performance in two-sample test scenarios.

**Major categories.** Currently, we identify that there are five main categories of SOTA SSL methods: consistency regularisation, pseudo-labelling, graph-based, generative models and hybrid (often a combination of consistency regularisation and pseudo-labelling) (Yang et al., 2023). We will succinctly explicate how they work, and how they are applied for our downstream two-sample testing tasks in the experiments of various levels of HDGM.

- *Consistency Regularisation:* Based on the manifold assumption or the smoothness assumption, the consistency regularisation methods apply consistency constraints to the final loss function, where the intuition is that if the data follows the smoothness assumption or manifold assumption, even though we construct some perturbations in the inputs, it will not influence the output of classification (Xie et al., 2020).

- *Pseudo-Labelling:* Pseudo-labelling uses its own predictions to generate labels for unlabeled data, which are then used to further train the model. It relies on the assumptions that model's high-confidence predictions are accurate. This assumption is based on the cluster assumption for the validity and efficacy of propagating labels to unlabelled data based on model predictions (Lee et al., 2013).

- *Graph-Based:* Graph-based methods will construct a similarity graph based on the raw dataset, where each node represents a data instance, and weighted-edge represents the similarity between two data instances. Based on the smoothness assumption, the label information can be propagated from labelled nodes to unlabelled nodes, if two nodes are closely connected in the constructed graph (Song et al., 2021b).

- *Generative Models:* Generative methods learn to model the underlying distribution of both labelled data and unlabelled data, using this learned representation to generate new data points and infer missing labels. Based on the manifold assumption, the generative models aim to learn the underlying low-dimensional manifold and generate data points that adhere to the same manifold, used for further model training (Kingma & Welling, 2013).

- *Hybrid:* Hybrid methods are just combination of multiple methods, such as consistency regularisation, pseudo-labelling, and sometimes generative approaches. These models typically rely on the smoothness assumption and cluster assumption, in order to infer the labels of unlabelled data (Sohn et al., 2020).

## C  EXPERIMENTAL DETAILS

### C.1  OVERVIEW OF DATASETS

**High-Dimensional Gaussian mixtures.** The *high dimensional Gaussian mixtures* (HDGM) benchmark is a synthetic dataset that is composed of multiple Gaussian distributions, each representing a cluster, which is proposed by Liu et al. (2020). In our experiments, we are considering bimodal Gaussian mixtures, which means the number of clusters remains 2 irrelevant to the dimension of the multivariate Gaussian distributions. In Section 4, we consider there are three levels of *HDGM*, which are *HDGM-Easy*, *HDGM-Medium* and *HDGM-Hard* in order to specify that most SOTA SSL methods are not suitable for two-sample testing problems. In other places rather than Section 4, we regard *HDGM* as *HDGM-Hard*. Under $H_0$, $\mathbb{P}$ and $\mathbb{Q}$ are the same, which denoted as *HDGM-S*; and under $H_1$, we slightly modify a mild covariance $\pm 0.5$ between first two dimensions in the covariance matrix of $\mathbb{Q}$ and other setups are the same as *HDGM-S*, which is referred to as *HDGM-D*. Thus, *HDGM-S* and *HDGM-D* are both noted by hard-level HDGM. The details of how to synthesize $\mathbb{P}$ and $\mathbb{Q}$ to derive *HDGM-Easy*, *HDGM-Medium*, *HDGM-Hard*, *HDGM-S* and *HDGM-D* are described in Table 3. We regard $n$ as the number of samples drawn from each cluster in each distribution and $N$ as the number of total samples drawn from both $\mathbb{P}$ and $\mathbb{Q}$, where $N = n \times c \times 2$. We conduct two experiments on *HDGM-D*, increasing the $N$ from $N = 1000$ to $N = 10000$ when keeping the dimension $d$ remain the same. One experiment is a low-dimensional *HDGM-D* with $d = 2$ and another is a high-dimensional *HDGM-D* with $d = 10$. Moreover, we conduct both low-dimensional and high-dimensional *HDGM-S* to show that the type-I error is controlled. The result is shown in Figure 5, which will be analyzed in the section 6.

**MNIST vs MNIST-Fake.** The *MNIST* datasets is a collection of 70,000 grayscale images of handwritten digits, ranging from 0 to 9, divided into a training set of 60,000 images and a test of 10,000 images (LeCun et al., 1998). The *MNIST*-Fake is the a set of 10,000 images generated by a pretrained *deep convolutional generative adversarial network* (DCGAN) (Radford et al., 2016). The MNIST benchmark (*MNIST* vs *MNIST*-Fake) is also proposed by Liu et al. (2020), aiming to test the performance of testing methods in the image space. Under $H_0$, we draw samples both from the *MNIST*-Fake. Under $H_1$, we compare the samples from real *MNIST*, $\mathbb{P}$, and samples from *MNIST*-Fake, $\mathbb{Q}$. We regard $N$ as the number of samples each drawn from $\mathbb{P}$ and $\mathbb{Q}$, where we

increase $N$ from $N = 200$ to $N = 1000$. The result of the average test power of all methods is displayed in the Table 2. All methods are tested with a reasonable type-I error rate.

**ImageNet vs ImageNet-Fake.** The *ImageNet* dataset is a comprehensive collection of over 14 million labeled high-resolution images belonging to roughly 22,000 categories (Deng et al., 2009). The *ImageNet*-Fake dataset comprises 10,000 high-quality images generated using the advanced *StyleGAN-XL* model, a state-of-the-art generative adversarial network designed for large and diverse datasets (Sauer et al., 2022). This benchmark (*ImageNet* vs *ImageNet*-Fake) extends the framework established by Liu et al. (2020) to a more complex and diverse image domain, testing the robustness of two-sample testing methods at a larger scale. Under the null hypothesis $H_0$, samples are drawn from *ImageNet*-Fake, while under the alternative hypothesis $H_1$, we compare samples from the real *ImageNet* dataset, $\mathbb{P}$, with those from *ImageNet*-Fake, $\mathbb{Q}$. We vary the number of samples drawn from each, $\mathbb{P}$ and $\mathbb{Q}$, from $N = 200$ to $N = 1000$ to examine the scalability of the test methods. The outcomes in terms of average test power across various methodologies are summarized in Table 2, with all tests maintaining a reasonable type-I error rate.

## C.2 IMPLEMENTATION DETAILS OF C2ST AND SSL-C2ST

- C2ST: a C2ST uses statistic in Eq. (1) to measure the difference of two samples. Rather than 3 phases described in the Algorithm 1, C2ST-based methods is purely supervised with only 2 phases. Implementation of C2ST paradigm is to only take *Phase 2* and *Phase 3* from Algorithm 1. Most of the implementation details are referenced from Lopez-Paz & Oquab (2018a) and Liu et al. (2020). The splitting portion of training and testing is always half to half, and the model architecture is the same for C2ST and SSL-C2ST, where first few layers are feature extractor and followed by a classification layer. Moreover, in the first step of *Phase 3*, we do not utilize the the softmax probability of the first value of the logits returned by the classifier to calculate the statistic of two samples, we apply Eq. (1) which directly derive the mean of the classification prediction accuracy of two samples.

- SSL-C2ST: a SSL version of C2ST. Most of the implementation details are described in the Algorithm 1, except we replace the way of calculating a statistic from Eq. (2) to Eq. (1).

In C2ST, we have a classifier $f$ consisting of a randomly initialized feature extractor $\phi_\theta(x)$ followed by a logistic regression layer with parameters $\boldsymbol{w}$ and $\boldsymbol{b}$, where

$$f(x) = \phi_\theta(x) \times \boldsymbol{w} + \boldsymbol{b}.$$

As the $f$ is a binary classifier, $f(x) = [z_0, z_1]$ and $\text{softmax}(f(x)) = [p_0, p_1]$, where $p_0 + p_1 = 1$. All parameters $\theta$, $\boldsymbol{w}$ and $\boldsymbol{b}$ are updated through the supervised learning on the training set, which aims to minimize the occurrence of incorrect predictions. Then, use the empirical probability of the correct predictions on an unseen testing set to measure the difference between two samples.

However, in SSL-C2ST, we have $g$ consisting of a feature extractor $\phi_a(x)$ trained on $S_P^{\text{tr}} \cup S_P^{\text{te}} \cup S_Q^{\text{tr}} \cup S_Q^{\text{te}}$ without labels via unsupervised learning and a logistic regression layer for subsequent supervised training purpose. In the unsupervised learning step, we use $\phi_a(x)$ to extract a latent feature vector $z$ from the input $x$, and then use a decoder $\phi_a^{-1}(x)$ to reconstruct $z$ to a reconstructed $x'$. We update the parameters of $\phi_a$ by minimizing the difference between the reconstructed input $x'$ and the original input $x$. After the unsupervised training procedure, we add a classification layer after $\phi_a$ to form a classifier $g$, and train the classification layer in the same way as the C2ST.

## C.3 DETAILS OF SSL-C2ST-M AND OTHER MMD BASED METHODS

We first introduce SSL-C2ST-M and compare the following state-of-the-art testing methods on two benchmark datasets:

- SSL-C2ST-M: An advanced SSL-C2ST-based method. Rather than using the prediction labels (0 or 1) to measure the test accuracy, we utilize MMD to calculate the differences between output features extracted from the SSL-C2ST. The output features could be the output of the hidden layer or the logits output of the classifier trained by the SSL-C2ST, as we discuss in Section 5.1.

- C2ST-M: a C2ST-based method that is the same as C2ST, except it uses the statistic in Eq. (2) to measure the absolute mean differences between the probability of the logits of two samples, as we discuss in the Section 2. In Liu et al. (2020), this method is also called C2ST-L where L refers to logit.

- MMD-D: MMD with a deep kernel (Liu et al., 2020); a state-of-the-art testing method in the supervised paradigm. MMD-D learns a deep kernel by directly maximizing the test power of MMD, leading to an increase in test power on the testing set.

- MMD-FUSE: a state-of-the-art testing method in the unsupervised paradigm. It fuses several MMD statistics based on the simple kernel of different combinations of hyperparameters into a new powerful statistic, then conducts a permutation test based on the fused statistic (Biggs et al., 2023).

### C.3.1 IMPLEMENTATION DETAILS OF SSL-C2ST-M

In the implementation of SSL-C2ST-M, the classifier is trained with the same way as how we do in the SSL-C2ST. However, SSL-C2ST-M is more flexible in the procedures of testing. For datasets whose input vector size is small in SSL-C2ST, such as HDGM, we use the absolute value of differences between the mean of $p_0$ of samples from $\mathbb{P}$ and that of samples from $\mathbb{Q}$. It measures the mean probability that samples will be classified label 0 by SSL-C2ST. For image datasets that have large input vector size, such as *MNIST*, we use the hidden-layer output of the classifier trained by the SSL-C2ST, whose input vector size is 100, to compute the MMD between the features extracted from two samples. For high-dimensional image datasets, the latent vector with a larger size can contain more useful information to measure the difference between two extracted features.

### C.4 DETAILS OF HDGM DATASETS

Table 3 displays the details of how HDGM datasets are generated (Liu et al., 2020). Different levels of HDGM datasets are first proposed in this paper, in order to show why SOTA SSL methods cannot be directly applied in the two-sample testing problem. The level of HDGM is differed from whether the data points are highly overlapping or whether the clusters within the same distribution are isolated. For the *HDGM-Easy*, $\Delta_\mu = 10$ and $\Delta_q = 5$. For the *HDGM-Medium*, $\Delta_\mu = 10$ and $\Delta_q = 0$. For the *HDGM-Hard*, $\Delta_\mu = 0.5$ and $\Delta_q = 0$.

Table 3: Details of how to synthesize $\mathbb{P}$ and $\mathbb{Q}$ in the experiments. Let $c = 2$ be the number of the clusters in each distribution, $d > 2$ be the dimension of multivariate normal distribution of each cluster. $(\boldsymbol{\mu}_1, \ldots, \boldsymbol{\mu}_c)$ is a set of d-dimensional mean vector $\boldsymbol{\mu}_i$ that specifies that mean of each dimension in the distribution, where $\boldsymbol{\mu}_1 = \mathbf{0}_d, \boldsymbol{\mu}_i = \boldsymbol{\mu}_{i-1} + \Delta_\mu \times \mathbf{1}_d$. $I_d$ is the $d \times d$ identity matrix, $\Delta_\mu$ is the cluster mean difference within the same distribution, and $\Delta_q$ is the mean difference between $\mathbb{P}$ and $\mathbb{Q}$. $\Delta_1 = 0.5, \Delta_2 = -0.5$, and $\boldsymbol{\Sigma}_i = \begin{pmatrix} 1 & \Delta_i & \mathbf{0}_{d-2} \\ \Delta_i & 1 & \mathbf{0}_{d-2} \\ \mathbf{0}_{d-2}^T & \mathbf{0}_{d-2}^T & I_{d-2} \end{pmatrix}$.

| Datasets | $\mathbb{P}$ | $\mathbb{Q}$ |
|---|---|---|
| *HDGM-S* | $\sum_{i=1}^c \mathcal{N}(\boldsymbol{\mu}_i, I_d)$ | $\sum_{i=1}^c \mathcal{N}(\boldsymbol{\mu}_i, I_d)$ |
| *HDGM-D* | $\sum_{i=1}^c \mathcal{N}(\boldsymbol{\mu}_i, I_d)$ | $\sum_{i=1}^c \mathcal{N}(\boldsymbol{\mu}_i + \Delta_q, \boldsymbol{\Sigma}_i)$ |

### C.5 DETAILS OF COMPUTING RESOURCES

The experiments of the work are conducted on three platforms. One platform is a Nvidia-4090 GPU PC with Pytorch framework. The second platform is a High-performance Computer cluster with lots of Nvidia-A100 GPU with Pytorch framework. The last platform is a Nvidia-4090 GPU Window Subsystem for Linux with Jax framework. The memory of three platforms are all over 16 GB. The storage of disk of three platforms are all over 512 GB.

## C.6 A DISCUSSION ABOUT SUPERVISED SEQUENTIAL TWO-SAMPLE TESTING AND APPLICABILITY OF SSL-C2ST

Supervised sequential two-sample testing represents another approach to utilizing testing data (Pandeva et al., 2022). In this framework, a classifier is trained to determine whether two samples from a single batch originate from the same distribution. Initially, batches are split and fed sequentially into the classifier as testing data. Batches that do not reject the null hypothesis are concatenated with previous batches and used as training data for the classifier, continuing until all batches are exhausted or a single batch rejects the null hypothesis. The sequential nature of the test emerges from the use of e-values, which are updated as more data becomes available, allowing for a dynamic assessment of the testing hypothesis. However, this method should not be directly compared to our method due to different problem settings and designs. Firstly, in sequential two-sample testing, data are split into several batches and tests are conducted on single, small batches. Conversely, in other supervised two-sample testing approaches, data are only split into two halves, creating a trade-off between the number of training and testing samples.

Furthermore, the design of our SSL-C2ST method is compatible with any other supervised two-sample testing framework, including sequential two-sample testing. As long as a proportion of data is used for testing, we can remove the labels from this testing data and concatenate it into the training data. This allows us to learn IRs through representation learning, followed by the original supervised two-sample testing framework.

## C.7 EXPERIMENT RESULT OF SEQUENTIAL TWO-SAMPLE TESTING

In this part, we will display the result of supervised sequential two-sample test that proposed by Pandeva et al. (2022) on the HDGM-Hard dataset, and compared the result with original C2ST and SSL-C2ST in our problem setting. We can find that even though this method can have a small increase on the test power over the original C2ST method, but have a large decrease to our method. The number of batches we choose is five, if we choose the number of batches to two, it is exactly similar as C2ST; if we choose the number of batches to a large number like ten, the test power will drop down, since the test data size will be too small. Thus, we decide five as the number of batches, and C2ST-Sequential(5) in the Table 4 represent the supervised sequential two-sample testing with the number of batches equal to five.

Table 4: Experiment results of test power of sequential two-sample testing with Batch5 over original C2ST and our propose SSL-C2ST on HDGM-hard dataset. $N$ is the total size of two samples inputed in 100 trials.

| Method | N=4000 | N=6000 | N=8000 | Avg. |
|---|---|---|---|---|
| C2ST-Sequential (5) | 0.32 | 0.57 | 0.79 | 0.56 |
| C2ST | 0.29 | 0.49 | 0.78 | 0.52 |
| SSL-C2ST | 0.50 | 0.81 | 0.99 | 0.77 |

## C.8 FUTURE WORK

Autoencoder is the basic representation learning algorithm we introduce to enhance our SSL-C2ST, we can also replace it to more advanced representation algorithms, such as semi-supervised *variational autoencoder* (VAE) (Kingma et al., 2014), $\beta$-VAE (Higgins et al., 2016), or other autoencoder-based representation learning algorithms (Tschannen et al., 2018b).

## C.9 REPRODUCIBILITY

All the reproducible code can be found in the anonymous link.

# D  THEORETICAL ANALYSIS

## D.1  PROOF OF THEOREM 5.1

*Proof.* Let $f' \in \mathcal{C}_\phi : \mathcal{X} \to \{0,1\}$ be the SSL-C2ST classifier model which has the same model architecture as C2ST. Recall from Eq. (1), the accuracy of $f'$ on the testing set $\mathcal{S}_{\text{te}}$ is

$$\hat{t} = \frac{1}{n_{\text{te}}} \sum_{(z_k, l_k) \in \mathcal{S}_{\text{te}}} \mathbb{I}\left[f'(z_k) = l_k\right],$$

where $n_{\text{te}} = |\mathcal{S}_{\text{te}}|$, then we have that

$$\Pr\left(\mathbb{I}\left[f'(z) = l\right] = \tau\right) = \begin{cases} p & \text{if } \tau = 1, \\ 1 - p & \text{if } \tau = 0. \end{cases}$$

**Lemma D.1.** *Under null hypothesis $H_0 : \mathbb{P} = \mathbb{Q}$, samples $S_P$ and $S_Q$ follows the same distribution, so $n_{\text{te}}\hat{t}$ is the sum of identically distributed Bernoulli random variables with a probability of random-guessing $p_{H_0} = \frac{1}{2}$, which follows a Binomial$(n_{\text{te}}, p_{H_0})$. For a large $n_{\text{te}}$ and using the central limit theorem, $\hat{t}$ will converge to a $\mathcal{N}\left(\frac{1}{2}, \frac{1}{4n_{\text{te}}}\right)$.*

**Lemma D.2.** *Under $H_1 : \mathbb{P} \neq \mathbb{Q}$, $n_{\text{te}}\hat{t}$ is the sum of Bernoulli random variables that may not be identically distributed. In that way, $n_{\text{te}}\hat{t}$ follows a Poisson Binomial distribution, which can be approximated by a Binomial$(n_{\text{te}}\bar{p}, n_{\text{te}}\bar{p}(1-\bar{p}))$, where $\bar{p} = n^{-1}\sum_{k=1}^{m+n} p_k$ (Ehm, 1991). For a large $n_{\text{te}}$, a central limit theorem holds that $\hat{t}$ will converge to a $\mathcal{N}\left(\bar{p}, \frac{\bar{p}(1-\bar{p})}{n_{\text{te}}}\right)$. Let $\bar{p} = 1 - \epsilon(\mathbb{P}, \mathbb{Q}; f')$, where $\epsilon(\mathbb{P}, \mathbb{Q}; f') \in \left(0, \frac{1}{2}\right)$ represent the inability of $f'$ on distinguishing between $\mathbb{P}$ and $\mathbb{Q}$, then $\hat{t} \sim \mathcal{N}\left(1 - \epsilon, n_{\text{te}}^{-1}(\epsilon - \epsilon^2)\right)$.*

Thus, the Type-II error is defined as the probability of failing to reject $H_0$, while $H_1$ is actually true. This occurs when the test statistic $\hat{t}$, which follows the distribution under $H_1$, does not exceed the critical threshold determined by the null distribution $H_0$ at a specified significance level $\alpha$. According to Lemma D.1, the threshold value $t_\alpha$ can be calculated as

$$t_\alpha = \mu + z_\alpha \times \sigma = \frac{1}{2} + \Phi^{-1}(1 - \alpha) \times \frac{1}{\sqrt{4n_{\text{te}}}},$$

combined with Lemma D.2, so the Type-II error is

$$\beta = \Pr_{T \sim \mathcal{N}\left(1 - \epsilon, n_{\text{te}}^{-1}(\epsilon - \epsilon^2)\right)}(T < t_\alpha) = \Pr_{T \sim \mathcal{N}\left(1 - \epsilon, n_{\text{te}}^{-1}(\epsilon - \epsilon^2)\right)}\left(T < \frac{1}{2} + \frac{\Phi^{-1}(1 - \alpha)}{\sqrt{4n_{\text{te}}}}\right)$$

$$= \Pr_{T' \sim \mathcal{N}\left(0, n_{\text{te}}^{-1}(\epsilon - \epsilon^2)\right)}\left(T' < \frac{\Phi^{-1}(1 - \alpha)}{\sqrt{4n_{\text{te}}}} + \epsilon - \frac{1}{2}\right)$$

$$= \Pr_{Z \sim \mathcal{N}(0,1)}\left(Z < \sqrt{\frac{n_{\text{te}}}{\epsilon - \epsilon^2}}\left(\frac{\Phi^{-1}(1 - \alpha)}{\sqrt{4n_{\text{te}}}} + \epsilon - \frac{1}{2}\right)\right)$$

$$= \Phi\left(\sqrt{\frac{n_{\text{te}}}{\epsilon - \epsilon^2}}\left(\frac{\Phi^{-1}(1 - \alpha)}{\sqrt{4n_{\text{te}}}} + \epsilon - \frac{1}{2}\right)\right)$$

$$= \Phi\left(\frac{\Phi^{-1}(1 - \alpha)/2 + \left(\epsilon - \frac{1}{2}\right)\sqrt{n_{\text{te}}}}{\sqrt{\epsilon - \epsilon^2}}\right).$$

Thus, the test power is

$$\pi(\alpha, n_{\text{te}}, \epsilon) = 1 - \beta = 1 - \Phi\left(\frac{\Phi^{-1}(1 - \alpha)/2 + \left(\epsilon - \frac{1}{2}\right)\sqrt{n_{\text{te}}}}{\sqrt{\epsilon - \epsilon^2}}\right) = \Phi\left(\frac{\left(\frac{1}{2} - \epsilon\right)\sqrt{n_{\text{te}}} - \Phi^{-1}(1 - \alpha)/2}{\sqrt{\epsilon - \epsilon^2}}\right).$$

As we know $\Phi^{-1}(1 - \alpha)/2$ is a constant, for a reasonably fixed large $n_{\text{te}}$, if we are trying to maximizing the test power, we are actually maximize the first term of numerator, which is

$$\mathcal{J}(\mathbb{P}, \mathbb{Q}; f') = \max_\epsilon \frac{\left(\frac{1}{2} - \epsilon(\mathbb{P}, \mathbb{Q}; f')\right)}{\sqrt{\epsilon(\mathbb{P}, \mathbb{Q}; f') - \epsilon(\mathbb{P}, \mathbb{Q}; f')^2}}, \quad \text{where } \epsilon(\mathbb{P}, \mathbb{Q}; f') \in \left(0, \frac{1}{2}\right)$$

This is equivalent to

$$
\begin{aligned}
\mathcal{J}(\mathbb{P}, \mathbb{Q}; f') &= \min_{\epsilon} \frac{\epsilon(\mathbb{P}, \mathbb{Q}; f')}{\sqrt{\epsilon(\mathbb{P}, \mathbb{Q}; f') - \epsilon(\mathbb{P}, \mathbb{Q}; f')^2}} \\
&= \min_{\epsilon} \frac{\sqrt{\epsilon(\mathbb{P}, \mathbb{Q}; f')}}{\sqrt{1 - \epsilon(\mathbb{P}, \mathbb{Q}; f')}} \\
&= \min_{\epsilon} \frac{\epsilon(\mathbb{P}, \mathbb{Q}; f')}{1 - \epsilon(\mathbb{P}, \mathbb{Q}; f')}
\end{aligned}
$$

The proof of equivalence can be found at the end of the proof. Since $\epsilon(\mathbb{P}, \mathbb{Q}; f') \in \left(0, \frac{1}{2}\right)$, it is clear to see that directly minimizing the $\epsilon(\mathbb{P}, \mathbb{Q}; f')$ will optimize the objectives of maximizing the test power.

Moreover, we will show that the Type-I error is also controlled, which is the probability of reject $H_0$, while $H_0$ is true:

$$
\begin{aligned}
\Pr_{T \sim \mathcal{N}\left(\frac{1}{2}, \frac{1}{4 n_{\text{te}}}\right)} (T > t_\alpha) &= \Pr_{T \sim \mathcal{N}\left(\frac{1}{2}, (4 n_{\text{te}})^{-1}\right)} \left(T > \frac{1}{2} + \frac{\Phi^{-1}(1 - \alpha)}{\sqrt{4 n_{\text{te}}}}\right) \\
&= \Pr_{T' \sim \mathcal{N}(0, (4 n_{\text{te}})^{-1})} \left(T' > \frac{\Phi^{-1}(1 - \alpha)}{\sqrt{4 n_{\text{te}}}}\right) \\
&= \Pr_{Z \sim \mathcal{N}(0,1)} \left(Z > \Phi^{-1}(1 - \alpha)\right) \\
&= 1 - \Pr_{Z \sim \mathcal{N}(0,1)} \left(Z < \Phi^{-1}(1 - \alpha)\right) \\
&= 1 - \Phi\left(\Phi^{-1}(1 - \alpha)\right) \\
&= \alpha,
\end{aligned}
$$

**Proof of equivalence.**

If we define $f(\epsilon) = \frac{1/2 - \epsilon}{\sqrt{\epsilon - \epsilon^2}}$, $g(\epsilon) = \frac{\epsilon}{\sqrt{\epsilon - \epsilon^2}}$ and $D(\epsilon) = \sqrt{\epsilon - \epsilon^2}$, where $\epsilon \in (0, \frac{1}{2})$. The equation $\max_{\epsilon} f(\epsilon) = \max_{\epsilon} \left(\frac{1/2}{D(\epsilon)} - g(\epsilon)\right)$ holds. It is clear to find that $f(\epsilon)$ and $\frac{1/2}{D(\epsilon)}$ are monotonically decreasing over the domain of $\epsilon$. Thus, only if $g(\epsilon)$ is monotonically increasing over the domain of $\epsilon$, the equation $\max_{\epsilon} \left(\frac{1/2}{D(\epsilon)} - g(\epsilon)\right) = \max_{\epsilon} \left(\frac{1/2}{D(\epsilon)}\right) - \min_{\epsilon}(g(\epsilon))$ holds. Firstly, let us calculate the derivative of $D(\epsilon) = \left(\epsilon - \epsilon^2\right)^{1/2}$ w.r.t $\epsilon$,

$$
\begin{aligned}
D'(\epsilon) &= \frac{1}{2(\epsilon - \epsilon^2)^{1/2}} \cdot (1 - \epsilon + (-\epsilon)) \\
&= \frac{1 - 2\epsilon}{2D(\epsilon)},
\end{aligned}
$$

then, we take the derivative of $g(\epsilon) = \frac{\epsilon}{D(\epsilon)}$ w.r.t $\epsilon$,

$$
\begin{aligned}
g'(\epsilon) &= \frac{1 \cdot D(\epsilon) - \epsilon \cdot D'(\epsilon)}{D(\epsilon)^2} = \frac{1}{D(\epsilon)^2} \cdot \frac{2D(\epsilon)^2 - \epsilon(1 - 2\epsilon)}{2D(\epsilon)} \\
&= \frac{1}{\epsilon(1 - \epsilon)} \cdot \frac{2(\epsilon - \epsilon^2) - \epsilon(1 - 2\epsilon)}{2D(\epsilon)} \\
&= \frac{\epsilon}{\epsilon(1 - \epsilon) \cdot 2\sqrt{\epsilon(1 - \epsilon)}} = \frac{1}{2(1 - \epsilon)\sqrt{\epsilon(1 - \epsilon)}}.
\end{aligned}
$$

We can find that over the domain of $\epsilon \in (0, \frac{1}{2})$, $g'(\epsilon) > 0$, which concludes the proof. The reason why deriving the objective to be equivalent to $\min_{\epsilon} \frac{\epsilon}{\sqrt{\epsilon - \epsilon^2}}$ is we can simplify it to $\min_{\epsilon} \sqrt{\frac{\epsilon}{(1 - \epsilon)}} = \min_{\epsilon} \frac{\epsilon}{(1 - \epsilon)}$, where $\epsilon \in (0, \frac{1}{2})$. In that way, it is quite straightforward to understand how minimizing $\epsilon$ can help to improve test power. $\qquad \square$

## D.2 PROOF OF THEOREM 5.3

Let $\epsilon(\mathbb{P}, \mathbb{Q}; f) \in \left(0, \frac{1}{2}\right)$ be the inability of $f$ to distinguish between distribution $\mathbb{P}$ and $\mathbb{Q}$. Then we define the $err_{\mathrm{te}}(f) = 2\epsilon(\mathbb{P}, \mathbb{Q}; f) \in (0, 1)$ to be the error rate of $f$ on distribution $\mathbb{P}$ and $\mathbb{Q}$.

**Theorem D.3.** *(Boucheron et al., 2000) Suppose function space $\mathcal{C} : \{f | f : \mathcal{X} \to \{0, 1\}\}$ has finite VC-dimension for $V \geq 1$. For any sample $\mathcal{S}$, any function $f$, we have*

$$\Pr\left[\sup_{f \in \mathcal{C}} |err_{\mathrm{te}}(f) - \widehat{err}_{\mathrm{te}}(f)| \geq \Delta\right] \leq 8\mathcal{C}[2m_{\mathrm{l}}, \mathcal{S}]e^{-m\Delta^2/8}.$$

*So for any $\Delta, \delta > 0$, if we draw from $\mathcal{S}$ a sample satisfying*

$$m_{\mathrm{l}} \geq \frac{8}{\Delta}\left(\ln(\mathcal{C}[m_{\mathrm{l}}, \mathcal{S}]) + \ln\left(\frac{8}{\delta}\right)\right),$$

*then, with probability at least $1 - \delta$, all functions $f$ satify $|err_{\mathrm{te}}(f) - \widehat{err}_{\mathrm{te}}(f)| \leq \Delta$.*

*Proof.* The given unlabelled sample size implies that with probability $1 - \delta/2$, all $f' \in \mathcal{C}$ have

$$|\widehat{err}_{\mathrm{unl}}(\phi_{f'}) - err_{\mathrm{unl}}(\phi_{f'})| \leq \sqrt{\frac{\ln\left(\frac{4s}{\delta}\right)}{2m_{\mathrm{u}}}} \leq \Delta,$$

which also implies that

$$\widehat{err}_{\mathrm{unl}}(\phi_{f'^*}) \leq err_{\mathrm{unl}}(\phi_{f'}) + \sqrt{\frac{\ln\left(\frac{4s}{\delta}\right)}{2m_{\mathrm{u}}}} \leq \xi + \sqrt{\frac{\ln\left(\frac{4s}{\delta}\right)}{2m_{\mathrm{u}}}} \leq \xi + \Delta.$$

Using the standard VC bounds (e.g., Theorem D.3), the labelled sample size $m_{\mathrm{l}}$ implies that with probability at least $1 - \delta/4$, all $f' \in \mathcal{C}_{\phi, \mathcal{S}, \chi}(\xi + 2\Delta)$ have $|err_{\mathrm{te}}(f) - \widehat{err}_{\mathrm{te}}(f)| \leq \Delta$. Then, by Hoeffding bounds, with probability at least $1 - \delta/4$ we have

$$\widehat{err}_{\mathrm{te}}(f'^*) \leq err_{\mathrm{te}}(f'^*) + \sqrt{\log(4/\delta)/2m_{\mathrm{l}}} \leq err_{\mathrm{te}}(f'^*) + \Delta.$$

Therefore, with probability at least $1 - \delta$, the $f' \in \mathcal{C}_\phi$ that optimizes $\widehat{err}_{\mathrm{te}}(f')$ subject to $\widehat{err}_{\mathrm{unl}}(\phi_{f'}) \leq \xi + \Delta$ has

$$\widehat{err}_{\mathrm{te}}(f') \leq err_{\mathrm{te}}(f'^*) + \sqrt{\frac{\ln\left(\frac{4s}{\delta}\right)}{2m_{\mathrm{u}}}} + \sqrt{\log(4/\delta)/2m_{\mathrm{l}}} \leq err_{\mathrm{te}}(f'^*) + \sqrt{\frac{\ln\left(\frac{4s}{\delta}\right)}{2m_{\mathrm{u}}}} + \Delta.$$

Moreover, since we have $\widehat{err}_{\mathrm{te}}(f') = \Pr_{(z_i, l_i) \sim \mathcal{S}}[f'(z_i) \neq l_i] \in (0, 1)$ which is proportional to the empirical inability $\hat{\epsilon}(S_P, S_Q; f') \in \left(0, \frac{1}{2}\right)$. Thus, we can conclude the following inequality

$$2\hat{\epsilon}(S_P, S_Q; f') \leq err_{\mathrm{te}}(f'^*) + \Delta + \sqrt{\frac{\ln\left(\frac{4s}{\delta}\right)}{2m_{\mathrm{u}}}},$$

since $err_{\mathrm{te}}(f'^*) = 2\epsilon(\mathbb{P}, \mathbb{Q}; f'^*)$,

$$\hat{\epsilon}(S_P, S_Q; f') \leq \epsilon(\mathbb{P}, \mathbb{Q}; f'^*) + \frac{\Delta}{2} + \sqrt{\frac{\ln\left(\frac{4s}{\delta}\right)}{8m_{\mathrm{u}}}},$$

which concludes the proof. $\qquad\square$

