# OpenReview forum: "Revisit Non-parametric Two-sample Testing as a Semi-supervised Learning Problem"
_ICLR.cc/2025/Conference — Submitted to ICLR 2025_

### Official Review · Reviewer_47NZ · 2024-10-29

**Soundness:** 3
**Presentation:** 3
**Contribution:** 3
**Rating:** 6
**Confidence:** 3

**Summary:**

This paper proposes a novel perspective to consider non-parametric two-sample testing as a semi-supervised learning (SSL) problem, introducing the SSL-based Classifier Two-Sample Test (SSL-C2ST) framework. The authors propose a two-step approach: first, learn IRs using all data, then fine-tune IRs with only labelled data to learn DRs, which can both utilize information from whole dataset and adapt the discriminative power to the given data.

**Strengths:**

1.The paper is written well and is easy to understand.

2. The studied problem is very important.

3. The results seem to outperform state-of-the-art.

**Weaknesses:**

1. My concern of the learning setting that is proposed in this paper is its difference and benefits compared to positive-unlabeled learning [1] and semi-supervised out-of-distribution detection [2]. Could the authors clarify or include some empirical justifications on the benefits of the proposed learning algorithm?

2. What are the key assumptions of the theoretical analysis made in the paper? Are there any empirical justification on their validity on large datasets?

[1] Garg et al., Mixture Proportion Estimation and PU Learning: A Modern Approach, NeurIPS 2021

[2] Du et al., How Does Unlabeled Data Provably Help Out-of-Distribution Detection? ICLR 2024

**Questions:**

see above

---

> ### Author Response · Authors · 2024-11-21
>
> Thanks for reading our paper and for your valuable feedbacks. We will answer your questions and concerns below.
>
> **Weakness 1:**
>
> > My concern of the learning setting that is proposed in this paper is its difference and benefits compared to positive-unlabeled learning [1] and semi-supervised out-of-distribution detection [2]. Could the authors clarify or include some empirical justifications on the benefits of the proposed learning algorithm?
>
> **Response 1:**
>
> We appreciate the reviewer’s question about the differences and benefits of the proposed learning setting compared to positive-unlabeled (PU) learning and semi-supervised out-of-distribution (OOD) detection. While we frame two-sample testing as a semi-supervised learning (SSL) problem, **we strictly follow the problem setting of two-sample testing,** where the goal is to determine whether two given samples are drawn from the same distribution.
>
> This setting makes it challenging to apply PU learning, where the focus is on having labeled data for one class and a large amount of unlabeled data. Similarly, it is hard to align with OOD detection, which focuses on monitoring whether individual data points during operation deviate from the training distribution, without requiring a second group of given samples for comparison. We will further distinguish our setting from both papers mentioned in your comments and cite both papers in discussion, which can help readers understand our paper clearly.
>
> Regarding the effectiveness of the proposed learning algorithm, we would like to address a common misunderstanding about our work. **Our aim is to frame two-sample testing as an SSL problem** to alleviate the trade-off inherent in the supervised paradigm, where information from unlabeled testing data cannot be fully utilized while maintaining discriminative power.
>
> With this motivation, we initially tested whether existing SSL techniques could be directly applied to two-sample testing but found they failed due to violated assumptions. This led us to explore **why these techniques fail** and propose a **viable method** that works specifically in the two-sample testing scenario. We found that those techiques fail because of violation of strict SSL assumptions both empirically and theoretically. Thus, we aim to provide a viable method that our data theoretically holds its assumption. Moreover, the **empirical effectiveness of our proposed method**—outperforming other two-sample testing methods—has been clearly demonstrated in the paper.
>
> Ultimately, our goal is to **pave the way for future research** by highlighting the research gap between two-sample testing and SSL, enabling the development of more advanced and adapted SSL techniques for this field. We hope this explanation clarifies the main contribution and the problem setting of our work.
>
> **Weakness 2:**
>
> > What are the key assumptions of the theoretical analysis made in the paper? Are there any empirical justification on their validity on large datasets?
>
> **Response 2:**
>
> As stated in the paper, our proposed pipeline adds an AE-based representation learning step to the original supervised paradigm. **This approach assumes that the data satisfies the manifold assumption that is high dimensional data lies approximately on a lower-dimensional manifold**, which is true for all kinds of two-sample testing data.
>
> Regarding empirical justification of the validity on large datasets, none of the ones we provide in our experiments are toy datasets. The HDGM-Hard with dimension 10 is already a challenging large dataset. Even our proposed outperforming method requires 10,000 input samples to reach a test power of 1, demonstrating its capability on large datasets. For the real-life image dataset (MNIST or Imagenet versus their FAKE generated images), both of the distributions contain over 50,000 images. However, since real-life data is generally easier to distinguish than statistically generated edge cases, our method converges to a test power of 1 with fewer than 1,000 drawn samples.
>
> These results validate the assumptions and effectiveness of our method across both large synthetic and real-world large datasets.
>
> **We would be delighted to engage in further discussions to ensure your concerns are fully addressed.**

---

> > ### Author Response · Authors · 2024-11-26
> >
> > Dear Reviewer 47NZ,
> >
> > We appreciate the time and effort that you have dedicated to reviewing our manuscript.
> >
> > We have carefully addressed all your queries. Could you kindly spare a moment to review our responses?
> >
> > Have our responses addressed your major concerns?
> >
> > If there is anything unclear, we will address it further. We look forward to your feedback.
> >
> > Best regards,
> >
> > The authors of Submission 6635

---

> > > ### Author Response · Authors · 2024-11-26
> > >
> > > Dear Reviewer 47NZ,
> > >
> > > Thank you again for your time and effort in reviewing our paper.
> > >
> > > We have carefully addressed all your comments and prepared detailed responses. Could you kindly review them at your earliest convenience?
> > >
> > > As the discussion period is coming to a close, we wanted to check if there are any remaining concerns preventing you from adjusting your score. If there is any additional clarification we can provide, please do not hesitate to let us know, and we will address it promptly.
> > >
> > > Best regards,
> > >
> > > Authors of Submission 6635

---

> > > > ### Author Response · Authors · 2024-11-29
> > > >
> > > > Dear Reviewer 47NZ,
> > > >
> > > > Thank you for your valuable feedback and constructive comments on our submission.
> > > >
> > > > Since we have already solve all the concerns that has been proposed, we would like to ask if you have any additional concerns or questions about the paper that we can address. If there are specific points that remain unclear, please feel free to point them out—we are more than willing to discuss and provide further explanations.
> > > >
> > > > If all your concerns have been addressed satisfactorily, we would sincerely appreciate it if you could consider revisiting your scores and generously increasing them, if appropriate, to reflect the clarified contributions of the paper.
> > > >
> > > > Thank you once again for your thoughtful review and for helping us improve our work.
> > > >
> > > > Best regards,
> > > >
> > > > Authors of Submission 6635

---

> > > > > ### Author Response · Authors · 2024-11-30
> > > > >
> > > > > Dear Reviewer 47NZ,
> > > > >
> > > > > We hope this message finds you well. As the closing date for the review process is approaching, we wanted to follow up to ensure that all of your concerns regarding our paper have been addressed. If there are any remaining questions or issues you would like us to clarify, we would be more than happy to discuss them further.
> > > > >
> > > > > If all your concerns have been resolved, we kindly ask if you could consider revisiting your scores and updating them to reflect the clarified contributions of our work. Your constructive feedback has been invaluable, and we sincerely appreciate the time and effort you have devoted to reviewing our paper.
> > > > >
> > > > > Thank you once again for your thoughtful evaluations, and we look forward to hearing from you.
> > > > >
> > > > > Best regards,
> > > > >
> > > > > Authors of Submission 6635

---

> > > > > > ### Author Response · Authors · 2024-12-01
> > > > > >
> > > > > > Dear Reviewer 47NZ,
> > > > > >
> > > > > > We hope this message finds you well. As the closing date for the review process is approaching, we wanted to follow up to ensure that all of your concerns regarding our paper have been addressed. If there are any remaining questions or issues you would like us to clarify, we would be more than happy to discuss them further.
> > > > > >
> > > > > > If all your concerns have been resolved, we kindly ask if you could consider revisiting your scores and updating them to reflect the clarified contributions of our work. Your constructive feedback has been invaluable, and we sincerely appreciate the time and effort you have devoted to reviewing our paper.
> > > > > >
> > > > > > Thank you once again for your thoughtful evaluations, and we look forward to hearing from you.
> > > > > >
> > > > > > Best regards,
> > > > > >
> > > > > > Authors of Submission 6635

---

> > > > > > > ### Author Response · Authors · 2024-12-02
> > > > > > >
> > > > > > > Dear Reviewer 47NZ,
> > > > > > >
> > > > > > > We hope this message finds you well. As the closing date for the review process is approaching, we wanted to follow up to ensure that all of your concerns regarding our paper have been addressed. If there are any remaining questions or issues you would like us to clarify, we would be more than happy to discuss them further.
> > > > > > >
> > > > > > > If all your concerns have been resolved, we kindly ask if you could consider revisiting your scores and updating them to reflect the clarified contributions of our work. Your constructive feedback has been invaluable, and we sincerely appreciate the time and effort you have devoted to reviewing our paper.
> > > > > > >
> > > > > > > Thank you once again for your thoughtful evaluations, and we look forward to hearing from you.
> > > > > > >
> > > > > > > Best regards,
> > > > > > >
> > > > > > > Authors of Submission 6635

---

### Official Review · Reviewer_4szV · 2024-11-02

**Soundness:** 2
**Presentation:** 1
**Contribution:** 1
**Rating:** 3
**Confidence:** 3

**Summary:**

This paper revisits non-parametric two-sample testing as a semi-supervised learning problem. Specifically, AE-based representation learning is used with labeled data by removing labels, and then the standard C2ST procedure is used with the labeled data. The superior test power was demonstrated on two image datasets, i.e., MNIST and ImageNet.

**Strengths:**

- The paper demonstrated that using unsupervised learning, i.e., AE-based representation learning, improved the test power of C2ST.

**Weaknesses:**

- The datasets used in experiments were from the computer vision domain only. It is not clear that the proposed method will be effective in other domains. At least, the datasets used in the existing study need to be used to demonstrate the usefulness of the proposed method.
- Although interesting insights into improving C2ST with unlabeled data exist, the current presentation is not easy to follow. For example, the theoretical analyses have some unclear points.

**Questions:**

- In Step 1, AE-based representation learning is adopted. Another possible method is recent self-supervised learning. Is AE the best choice?
- Since Theorem 5.1 cites Lopez-Paz & Oquab (2018) and Theorem D.3 (the proof of Theorem 5.3) cites Boucheron et al. (2000) at the beginning, it is not easy to distinguish which parts are the contribution of this paper. Could you clarify it?
- $N = n$?, $N = m$?, or $N = n+m$?
- It is unclear how many unlabeled samples help improve the test power.
- What are $\delta$ and $\delta_{m_u,m_l}$ in Theorem 5.3?
- Semi-supervised learning often assumes labeled data and (a large amount of) unlabeled data, and both labeled and unlabeled data are simultaneously involved in training. However, this paper assumes labeled data only. In this sense, the paper sounds like it claims that an unsupervised pre-training step (by removing labels of labeled data) helps improve C2ST performance.
- From the SSL setting viewpoint, the proposed method can be compared with SSL methods using, e.g., 1,000 labeled and 10,000 unlabeled samples.
- In Sec 5.2, the setting in this paper implies m_l=m_u and labeled samples are identical to unlabeled samples. Applying the existing analysis to SSL may not be valid since the assumptions of the existing analysis and this paper are probably different. This paper should carefully discuss such a difference in the assumption.

---

> ### Author Response · Authors · 2024-11-21
>
> Thanks for reading our paper and for your valuable feedbacks. We will answer your questions and concerns below.
>
> **Weakness 1:**
>
> > The datasets used in experiments were from the computer vision domain only. It is not clear that the proposed method will be effective in other domains. At least, the datasets used in the existing study need to be used to demonstrate the usefulness of the proposed method.
>
> **Response 1:**
>
> We would like to address the concern regarding the datasets used in our experiments. Firstly, our **main experiments were conducted on the HDGM datasets**, which are widely recognized as a **primary benchmark in the two-sample testing scenario.** The image datasets, such as MNIST vs. FAKE-MNIST and Imagenet vs. FAKE-Imagenet, were included to demonstrate the **practical applicability** of two-sample testing methods in real-world scenarios. This combination of synthetic benchmarks and real-life applications ensures a comprehensive evaluation of the empirical power of our proposed approach.
>
> Furthermore, we would like to clarify a potential misunderstanding. **The primary focus of this paper is not solely on proposing a new two-sample testing method.** Instead, we aim to **pave the way for a novel research direction** by addressing two key research questions:
>
> 1. By intuition, can we directly frame two-sample testing as a semi-supervised learning (SSL) problem?
> 2. If not, what are the challenges, and how can we propose a viable approach for further research?
>
> Our contribution lies in being the first to propose this perspective and providing insights into how SSL can address the trade-offs and limitations in existing two-sample testing methods.
>
> We hope this clarification addresses the concern and highlights the significance of our research questions in advancing the field.

---

> > ### Author Response · Authors · 2024-11-21
> >
> > **Weakness 2:**
> >
> > > Although interesting insights into improving C2ST with unlabeled data exist, the current presentation is not easy to follow. For example, the theoretical analyses have some unclear points.
> >
> > **Response 2:**
> >
> > We appreciate the reviewer's feedback regarding the theoretical analyses and would like to clarify any points of confusion. However, it is not clear which specific aspects are considered unclear, and we would be happy to provide further explanation as needed.
> >
> > Firstly, the **SSL theoretical analysis for the discriminative model is referenced from [1]**, where we adapt the framework specifically to our proposed method. In our approach, we learn the function from a **smaller function subspace C_ϕ**, as ϕ is learned from unlabeled data before learning the final model.
> >
> > Moreover, compared to purely supervised methods, the theoretical results in our paper demonstrate that a **large enough unlabeled sample size decreases the upper bound of the empirical error.** This reduction in empirical error translates into an **increase in the lower bound of the test power** due to the relationship between ϵ (error) and test power. In the context of two-sample testing, under the alternative hypothesis, the **test power will always converge to 1 as the sample size increases.** Therefore, by increasing the lower bound of the test power, our method ensures higher test power with fewer samples compared to supervised methods.
> >
> > We hope this explanation provides additional clarity regarding the theoretical analyses. Please let us know if there are specific areas where further elaboration is needed.
> >
> > [1] Maria-Florina Balcan and Avrim Blum. A discriminative model for semi-supervised learning. Journal of the ACM, 57(3):19:1–19:46, 2010

---

> > > ### Author Response · Authors · 2024-11-21
> > >
> > > **Question 1:**
> > >
> > > > In Step 1, AE-based representation learning is adopted. Another possible method is recent self-supervised learning. Is AE the best choice?
> > >
> > > **Response 1:**
> > >
> > > We appreciate the reviewer’s question regarding whether AE-based representation learning is the best choice for Step 1. As we mentioned in addressing our first research question, **we indeed tested whether pseudo-labeling (a common self-supervised learning technique) is viable.** However, our experiments demonstrated that due to the nature of the datasets in two-sample testing, which **highly violate the assumptions of pseudo-labeling methods**, these techniques cannot be directly applied to this scenario.
> > >
> > > That said, we are currently exploring ways to adapt such methods for two-sample testing, as we believe self-supervised learning techniques hold potential when appropriately tailored. However, **this exploration is beyond the main focus of this work.** In this paper, our primary contribution is to demonstrate that **framing two-sample testing as a semi-supervised learning (SSL) problem is both worthy and viable.** The AE-based approach serves as a simple starting point to address this perspective, paving the way for future research to adapt and advance other methods, including self-supervised learning, to the two-sample testing context.
> > >
> > > We hope this explanation clarifies the rationale for our choice and the scope of this work.
> > >
> > > **Question 2:**
> > >
> > > > Since Theorem 5.1 cites Lopez-Paz & Oquab (2018) and Theorem D.3 (the proof of Theorem 5.3) cites Boucheron et al. (2000) at the beginning, it is not easy to distinguish which parts are the contribution of this paper. Could you clarify it?
> > >
> > > **Response 2:**
> > >
> > > We appreciate the reviewer’s question regarding the contributions of Theorem 5.1 and Theorem 5.3. While both theorems build on foundational work from previous studies, **our theoretical contributions lie in adapting and applying the SSL results to the two-sample testing context.**
> > >
> > > Specifically, for **Theorem 5.1**, although it is referenced from Lopez-Paz & Oquab (2018), the original focus of the theorem was on the **effect size of the C2ST model in the field of two-sample testing.** In our work, we adapt this theorem to focus on the **error rate of the model**, which is critical to evaluating the performance of our proposed SSL framework.
> > >
> > > For **Theorem 5.3**, while it originates from Boucheron et al. (2000) and Balcan et al. (2010), and discusses the **error rate of semi-supervised discriminative models**, our contribution lies in **validating that this semi-supervised learning theorem can be applied to the two-sample testing scenario**. We achieve this by connecting the error term in the SSL framework to the test power in two-sample testing, providing a verification on how SSL can indeed increase the test power of purely supervised two-sample testing methods.
> > >
> > > We hope this clarifies how these theorems were adapted and extended to support the main contributions of this paper. Please let us know if further elaboration is needed.

---

> > > > ### Author Response · Authors · 2024-11-21
> > > >
> > > > **Question 3:**
> > > >
> > > > > N=n?, N=m?, or N=n+m?
> > > >
> > > > **Response 3:**
> > > >
> > > > We apologize for the misunderstanding regarding the notation **N** in the experimental results. To clarify:
> > > >
> > > > - In **Table 2**, as we specified, **N = m = n**, where mmm and nnn represent the sample sizes for each of the two distributions being tested.
> > > > - In **Figure 5**, however, **N = m + n**, which represents the total sample size received for the experiment.
> > > >
> > > > While we have explained these two meanings of **N** in **Appendix C.1 on Page 15**, we recognize that this inconsistency may still cause confusion. To improve clarity and ensure easy understanding, we will make the usage of **N** consistent throughout the revised version of the paper.
> > > >
> > > > We apologize for any confusion caused by this oversight and appreciate your understanding.
> > > >
> > > > **Question 4:**
> > > >
> > > > > It is unclear how many unlabeled samples help improve the test power.
> > > >
> > > > **Response 4:**
> > > >
> > > > We would like to clarify that the relationship between the number of unlabeled samples and the improvement in test power is addressed in the paper. **As stated on line 425 of Page 8,** to reach an arbitrarily small value Δ, **a sufficiently large number of unlabeled samples $m_u$ is required** to improve the test power.
> > > >
> > > > This theoretical insight highlights that the amount of unlabeled data plays a crucial role in reducing the error bound and subsequently increasing the test power.
> > > >
> > > > We hope this explanation clarifies the point, but we are happy to provide further elaboration if needed.

---

> > > > > ### Author Response · Authors · 2024-11-21
> > > > >
> > > > > **Question 5:**
> > > > >
> > > > > > What are δ and δmu,ml in Theorem 5.3?
> > > > >
> > > > > **Response 5:**
> > > > >
> > > > > As stated in Theorem 5.3, Δand $\Delta_{m_u, m_l}$ are inspired by the learning theory of semi-supervised learning. **Δ can be regarded similar to an error term in PAC learning theory,** where achieving a smaller error requires a larger sample size.
> > > > >
> > > > > In the context of semi-supervised learning, this error term is influenced by both the size of the unlabeled $m_u$ and labeled $m_l$ data. To reflect this dependency, we refine the notation to $\Delta_{m_u, m_l}$, which represents the achievable error as a function of these two sample sizes. **The theorem demonstrates that with sufficiently large sample sizes for both labeled and unlabeled data, a smaller error can be achieved.**
> > > > >
> > > > > We hope this clarification resolves any confusion, but we are happy to provide further explanation if needed.
> > > > >
> > > > > **Question 6:**
> > > > >
> > > > > > Semi-supervised learning often assumes labeled data and (a large amount of) unlabeled data, and both labeled and unlabeled data are simultaneously involved in training. However, this paper assumes labeled data only. In this sense, the paper sounds like it claims that an unsupervised pre-training step (by removing labels of labeled data) helps improve C2ST performance.
> > > > >
> > > > > **Response 6:**
> > > > >
> > > > > We acknowledge that current SSL methods often assume both labeled and unlabeled data are simultaneously involved in training, as you correctly pointed out. However, as we stated in our problem setting, the scenario of **two-sample testing differs significantly**.
> > > > >
> > > > > In two-sample testing, we are given two samples, $S_P$ and $S_Q$, and aim to determine whether they originate from the same distribution. In the purely supervised paradigm, data splitting is required to perform permutation testing on the testing set. However, this approach **discards the inherent information from the testing set**, which could otherwise enhance the ability for representation learning. To address this limitation, we propose a **viable way to utilize this discarded information** from the testing set.
> > > > >
> > > > > It is important to emphasize, as we have stated several times in the paper, that our goal is **not to propose a SOTA SSL method.** Instead, we aim to test whether the discarded information from the unlabeled testing data in two-sample testing can be effectively utilized. By the definition of SSL—where leveraging the information from P(x) (the distribution of data) to improve the inference of P(y|x) qualifies as semi-supervised learning—our approach fits this framework. Specifically, we remove the labels of the labeled data and concatenate them with the unlabeled testing data for representation learning, so we are not just using the labeled data only. This approach is **transductive learning,** as mentioned on **line 198 on Page 4.**
> > > > >
> > > > > We also tested the implementation of SOTA SSL methods that can simultaneously use unlabeled testing data and labeled training data. However, due to the **violations of the smoothness and cluster assumptions** in two-sample testing scenarios, training under these assumptions led to **worse performance.**
> > > > >
> > > > > We hope this explanation clarifies the novelty and scope of our work in utilizing unlabeled testing data in two-sample testing scenarios.

---

> > > ### Comment · Reviewer_4szV · 2024-11-22
> > >
> > > Since $\phi_f$ is the feature extractor, $err_\mathrm{unl}(\phi_f)=1-\chi(f, \mathcal{S})$ is not clear. There seems to be a gap in applying the existing analysis [1] to the proposed method because unlabeled data is used for learning auto-encoder rather than a classifier. I probably missed something important to understand the analysis.
> > >
> > > [1] Maria-Florina Balcan and Avrim Blum. A discriminative model for semi-supervised learning. Journal of the ACM, 57(3):19:1–19:46, 2010

---

> > > > ### Author Response · Authors · 2024-11-23
> > > >
> > > > Thank you for your quick response, and we’d like to clarify this point further.
> > > >
> > > > Firstly, regarding the use of unlabeled data for learning an autoencoder rather than a classifier, we want to highlight that in the original C2ST model, the classifier architecture consists of **two parts**: the **first few layers act as a feature extractor**, and the subsequent layers as a classification layer. In our pipeline, we treat the first few layers as the feature extractor capable of obtaining IRs. By using unlabeled data to effectively pre-train and improve these initial layers, we enable the feature extractor to learn better representations, which are then combined with labeled data to further train the classification layers. Thus, while unlabeled data is not directly used to train the classification layers, it plays a critical role in **training the feature extractor layers** and, indirectly, the entire classifier model.
> > > >
> > > > Moreover, as defined in [1], **compatibility** is generally linked to the unlabeled error rate (unsupervised loss) in semi-supervised learning. Since our pipeline does not utilize unlabeled data to train the classification layer, we specify this definition to our case by focusing on the ability of the feature extractor (the first few layers of the classifier) to learn effective representations. Specifically, we redefine compatibility and the unlabeled error rate to be the unlabeled reconstruction error rate (unsupervised loss) in terms of the performance of the feature extractor within the classifier model.
> > > >
> > > > **This redefined unlabeled error rate is consistent with the original theoretical framework in [1],** as *"these bounds hold only for algorithms of a specific type (that first use the unlabeled data to choose a small set of “representative” hypotheses and then choose among the representatives based on the labeled data)"*. In our case, the autoencoder-based feature extractor uses the unlabeled data to learn IRs that align with the manifold assumption, which are then used to improve downstream classification tasks. The key insight remains that by leveraging unlabeled data to learn an effective ϕ, our method enables learning from a smaller, more optimal function space $C_\phi$, which significantly improves over learning f from the larger space C. By redefining the error rate in terms of reconstruction, we remain within the framework’s scope, as the unsupervised loss contributes directly to improving compatibility and reducing error in the subsequent classification step.
> > > >
> > > > If it will make readers confusing, we are open to revising the theoretical in the revised version to align with the original inductive learning setting and general unlabeled error rate. This will preserve the generality of the analysis while continuing to demonstrate the increased power of our pipeline.
> > > >
> > > > We hope this explanation resolves any remaining confusion, and we are happy to make further clarifications or adjustments if needed.

---

> ### Author Response · Authors · 2024-11-21
>
> **Question 7:**
>
> > From the SSL setting viewpoint, the proposed method can be compared with SSL methods using, e.g., 1,000 labeled and 10,000 unlabeled samples.
>
> **Response 7:**
>
> We appreciate the reviewer’s suggestion regarding comparisons under the SSL setting with configurations such as few labeled samples and lots of unlabeled samples. However, we would like to clarify that the **two-sample testing scenario differs fundamentally from the traditional SSL setting.**
>
> In two-sample testing, we can assume that **all input data is labeled** because the goal is to determine whether two given samples, $S_P$ and $S_Q$, are from the same distribution. The challenge lies in the **trade-off of splitting proportions** between training and testing sets, as also required by the supervised paradigm. Moreover, this splitting proportion will not influence the representation learning (we use all the input data for representation learning).
>
> While the trade-off of splitting proportions is a central issue in two-sample testing, it is **not the primary focus of our work.** Instead, we aim to explore whether and how the information from the testing set (treated as unlabeled data) can be utilized to enhance test performance. This conceptual shift is what we aim to demonstrate as viable and valuable in this work. We hope this explanation clarifies why traditional SSL data configurations are not directly applicable in the context of two-sample testing and how our work approaches this unique challenge.
>
> **Question 8:**
>
> > In Sec 5.2, the setting in this paper implies m_l=m_u and labeled samples are identical to unlabeled samples. Applying the existing analysis to SSL may not be valid since the assumptions of the existing analysis and this paper are probably different. This paper should carefully discuss such a difference in the assumption.
>
> **Response 8:**
>
> We would like to clarify that **$m_l$ and $m_u$ are not identical** in our setting. The unlabeled sample size $m_u$ must be **greater than $m_l$**, as it includes all the training data (with labels removed) and all of the testing data.
>
> Given this, we believe there is no additional assumption required to discuss, as our setting aligns with the framework used in the analysis.
>
> **We would be delighted to engage in further discussions to ensure your concerns are fully addressed.**

---

> > ### Comment · Reviewer_4szV · 2024-11-22
> >
> > Thank you for the clarification. Where is the information about the size of training and testing sets?

---

> > > ### Author Response · Authors · 2024-11-23
> > >
> > > Thank you for your response. Indeed, we did not explicitly specify the train-test splitting strategy in the experimental implementation section. However, we followed the convention in the previous supervised paradigm, which is a 50-50 split, and we will restate this detail in the experimental details in the appendix for better clarity.
> > >
> > > Regarding the relationship between the **unlabeled sample size** and the **labeled sample size**, this can be inferred from Figure 3 and Algorithm 1. As described, we use all the given samples (removing their labels) for representation learning, then perform the train-test split. The training set is subsequently used to train the classification layer, and the entire classifier model is used for permutation testing. Thus, the size of the whole given sample must necessarily be larger than the size of the training set, which is a subset derived from the entire sample.
> > >
> > > We will ensure these points are made clearer in the revised paper.

---

> > > > ### Comment · Reviewer_4szV · 2024-11-25
> > > >
> > > > > Specifically, we redefine compatibility and the unlabeled error rate to be the unlabeled reconstruction error rate (unsupervised loss) in terms of the performance of the feature extractor within the classifier model.
> > > >
> > > > Thank you for your answers.
> > > > In $err_\mathrm{unl}(\phi_f)=1-\chi(f, \mathcal{S})$, $\phi_f$ does not appear in the right-hand side. Probably, the authors may want to define, e.g., $\tilde{\chi}(\phi_f, \mathcal{S})$ and $\tilde{\chi}(\phi_f, x)$ to measure compatibility since $\chi(f, x)$ cannot be used with $\phi_f\colon \mathcal{X}\to\mathbb{R}^k$. After that, other terms in the theoretical analysis will also be corrected if necessary. Currently, a certain idea of the analysis is provided, but it is not enough to evaluate the validity of the analysis owing to unclear notations.
> > > >
> > > > > Indeed, we did not explicitly specify the train-test splitting strategy in the experimental implementation section. However, we followed the convention in the previous supervised paradigm, which is a 50-50 split, and we will restate this detail in the experimental details in the appendix for better clarity.
> > > >
> > > > Thank you. The information, a 50-50 split, is important for reproducing the results in this paper.

---

> > > > > ### Author Response · Authors · 2024-11-26
> > > > >
> > > > > Thank you for your thoughtful response. We appreciate your attention to the definition of $\text{err}_{\text{unl}}(\phi_f)$ and its relation to $\chi(f, S)$.
> > > > >
> > > > > It is indeed more formal to express $\text{err}_{\text{unl}}(\phi_f) = \chi(\phi_f, \mathcal{X})$. However, we would like to clarify that $\phi_f$ is inherently part of $f$, so it is not strictly correct to state that $\phi_f$ does not appear in the right-hand side of the equation. This connection aligns with our theoretical framework and ensures that the subsequent theoretical analysis is not impacted by this notation.
> > > > >
> > > > > In both the original referenced work and our proposed method, $\text{err}_{\text{unl}}$ is consistently used to represent the **empirical average**, while $\chi$ serves as a **theoretical definition** that measures how well the model fits the unlabeled dataset. Although we have redefined the compatibility in our specific case, this redefinition introduces only minor notational adjustments and does not affect the **main contributions or validity** of the theoretical analysis.
> > > > >
> > > > > Moreover, we explicitly state the definition of  $\chi(f, S)$ as the compatibility of how the feature extractor of $f$ fits the dataset $S$. Based on this definition, we believe the empirical unlabeled error, $\text{err}_{\text{unl}}(\phi_f)$, is clearly stated and reasonable.
> > > > >
> > > > > We hope this explanation clarifies our reasoning and helps you better understand our notation. Please let us know if further clarifications are needed.
> > > > >
> > > > > Best,
> > > > > The authors of Submission 6635

---

> > > > > > ### Comment · Reviewer_4szV · 2024-11-26
> > > > > >
> > > > > > Thank you for your explanaiton. However, it is still unclear. The feature extractor is defined as $\phi\colon\mathcal{X}\to\mathbb{R}^k$. The relation to $f$ cannot be found in the definition. Probably, there is another definition for the feature extractor $\phi_f$ different from the definition of the feature extractor $\phi$. If there is an equation showing "$\phi_f$ is inherently part of $f$," it would be helpful.

---

> > > > > > > ### Author Response · Authors · 2024-11-26
> > > > > > >
> > > > > > > Yes, we indeed define the relationship between the $\phi$ and $f$ in line 321 on Page 6. Thus, if this can clarify your concerns, we hope to receive an updated rating. Thanks.

---

> > > > > > > > ### Author Response · Authors · 2024-11-29
> > > > > > > >
> > > > > > > > Dear Reviewer 4szV,
> > > > > > > >
> > > > > > > > Thank you for your valuable feedback and constructive comments on our submission.
> > > > > > > >
> > > > > > > > Since we have already solve all the concerns that has been proposed, we would like to ask if you have any additional concerns or questions about the paper that we can address. If there are specific points that remain unclear, please feel free to point them out—we are more than willing to discuss and provide further explanations.
> > > > > > > >
> > > > > > > > If all your concerns have been addressed satisfactorily, we would sincerely appreciate it if you could consider revisiting your scores and generously increasing them, if appropriate, to reflect the clarified contributions of the paper.
> > > > > > > >
> > > > > > > > Thank you once again for your thoughtful review and for helping us improve our work.
> > > > > > > >
> > > > > > > > Best regards,
> > > > > > > >
> > > > > > > > Authors of Submission 6635

---

> > > > > > > > > ### Comment · Reviewer_4szV · 2024-11-30
> > > > > > > > >
> > > > > > > > > Thank you for your response. If $\chi(f, S)$ for $\phi_f$ is the authors' intention, Theorem 5.3 in this paper is Theorem 10 in [1]. The error term in Theorem 5.3 is replaced with the variant of the error term; the replaced error is proportional to the error term in the original theorem. The feature extractor is not explicitly handled in the analysis. If concrete examples of the compatibility function for the proposed algorithm are shown, it would be helpful to understand this paper since [1] showed the compatibility function for, e.g., graph-based method and co-training, but how to apply it to AE-based SSL is not mentioned. From this viewpoint, the novelty of the analysis is marginal.
> > > > > > > > >
> > > > > > > > > [1] Maria-Florina Balcan and Avrim Blum. A discriminative model for semi-supervised learning. Journal of the ACM, 57(3):19:1–19:46, 2010

---

> > > > > > > > > > ### Author Response · Authors · 2024-11-30
> > > > > > > > > >
> > > > > > > > > > Thank you for further sharing your concerns and providing additional feedback. We greatly appreciate your thoughtful comments.
> > > > > > > > > >
> > > > > > > > > > To address your points:
> > > > > > > > > >
> > > > > > > > > > * Compatibility Function: Compatibility is a notation used to represent how well the model fits the unlabeled data. In our work, we have redefined compatibility to specifically measure how the feature extractor of the model fits the unlabeled data. If a concrete example of the incompatibility function is needed, it can be found in the objective function in Equation 3, as this is how we estimate the unlabeled error in our framework.
> > > > > > > > > >
> > > > > > > > > > * Novelty of the Analysis: We acknowledge that our theoretical analysis closely follows existing work in semi-supervised learning. However, the purpose of this analysis is not to advance the theoretical understanding of SSL itself but to demonstrate the validity of using unlabeled data to improve the test power of two-sample testing methods. Our goal is to pave the way for future research to develop advanced and adapted SSL techniques for two-sample testing scenarios. We see our work as an important step toward bridging the gap between two fields, providing a foundation for further exploration.
> > > > > > > > > >
> > > > > > > > > > We hope this explanation helps clarify our intentions and contributions. Please feel free to share any further questions or concerns—we would be happy to address them.

---

> > > > > > > > > > > ### Author Response · Authors · 2024-12-01
> > > > > > > > > > >
> > > > > > > > > > > Dear Reviewer 4szV,
> > > > > > > > > > >
> > > > > > > > > > > We hope this message finds you well. As the closing date for the review process is approaching, we wanted to follow up to ensure that all of your concerns regarding our paper have been addressed. If there are any remaining questions or issues you would like us to clarify, we would be more than happy to discuss them further.
> > > > > > > > > > >
> > > > > > > > > > > If all your concerns have been resolved, we kindly ask if you could consider revisiting your scores and updating them to reflect the clarified contributions of our work. Your constructive feedback has been invaluable, and we sincerely appreciate the time and effort you have devoted to reviewing our paper.
> > > > > > > > > > >
> > > > > > > > > > > Thank you once again for your thoughtful evaluations, and we look forward to hearing from you.
> > > > > > > > > > >
> > > > > > > > > > > Best regards,
> > > > > > > > > > >
> > > > > > > > > > > Authors of Submission 6635

---

> > > > > > > > > > > ### Comment · Reviewer_4szV · 2024-12-01
> > > > > > > > > > >
> > > > > > > > > > > Thank you for your answer. The equation (3) contains $\psi$, but $\chi(f, S)$ does not.

---

> ### Author Response · Authors · 2024-12-01
>
> Dear Reviewer 4szV,
>
> Thanks for proposing your another concern. Since $\psi$ is irrelevant to the f, we think there is no need to include $\psi$ into the compatibility notation. Just same as other semi-supervised learning techniques, like consistency regularisation, it will only show that the unsupervised loss to be $\frac{1}{N} \sum_{i=1}^N \|f(x_i) - f(a(x_i))\|^2$, since the augmentation function $a$ is irrelevant to the f, so there is no need to specify how $a$ related to the compatibility. Hope this addresses your concerns, if all your concerns have been resolved, we kindly ask if you could consider revisiting your scores and updating them to reflect the clarified contributions of our work.
>
> Best regards,
>
> Authors of Submission 6635

---

> > ### Comment · Reviewer_4szV · 2024-12-02
> >
> > Thank you for the clarification.
> >
> > - It would be great if the citation is provided for "like consistency regularisation, it will only show that the unsupervised loss to be $\frac{1}{N} \sum_{i=1}^N |f(x_i) - f(a(x_i))|^2$, since the augmentation function $a$ is irrelevant to the f, so there is no need to specify how $a$ related to the compatibility."
> >
> > - If Equation (3) is the compatibility function, it is better to mention the scale such that the range of Equation (3) becomes $[0, 1]$.
> >
> > - $\psi$ is learned in the proposed method but does not appear in the theoretical analysis. Since $\psi$ has parameters, it is often the case that the analysis would have the model complexity, such as the VC dimension, for $\psi$. Without $\psi$ in the analysis, it is unclear why the presented analysis is valid for the proposed method that involves representation learning through $\phi$ and $\psi$. $m_u$ in the current analysis is irrelevant to $\psi$, but $m_u$ would have dependency on the complexity of $\psi$. For example, if the VC dimension of $\psi$ is large, the reasonable analysis would suggest that we need a lot of unlabeled samples. Thus, the answer is unclear: " there is no need to include $\psi$ into the compatibility notation."

---

> > > ### Author Response · Authors · 2024-12-03
> > >
> > > Thanks for proposing your further concerns.
> > > * There are lots of citations for the consistency regularisation techniques in the SSL field, like [1], [2], [3]. In thier works, the unsupervised loss are all consist of the distance between $f(x)$ and $f(\tilde{x}) = f(g(x))$, where $g$ is any perturbation / augmentation function. Thus, including a function that is irrelevant to the $f$ in the unsupervised loss will not influence the general compatibility function $\chi(f, S)$ definition.
> > >
> > > * Indeed, we will revise it to the scaling of MSE to ensure that the range of unsupervised loss to be [0, 1]. Currently, due to the good fit of two-sample testing data to the manifold assumption, the empirical unsupervised loss are very close to 0 among all the datasets, we will make sure the theoretical analysis is rigorous.
> > >
> > > * Firstly, compatibility is a general definition used in the analysis of the discriminative model $f$, so irrelevant part $\psi$ is not included in the definition. Just as how generative model helps SSL, it trains the generative model $g$ on the unlabeled data and labeled data in order to synthesize extra training data to ensure the robustness of final classification model $f$. We can find $g$ is learned and irrelevant with the discrimative model $f$, and it is also not included in the general definition of the compatibility.
> > >
> > > Hope this addresses your concerns, if all your concerns have been resolved, we kindly ask if you could consider revisiting your scores and updating them to reflect the clarified contributions of our work.
> > >
> > > Best,
> > >
> > > Authors of Submission 6635
> > >
> > > [1] Laine S, Aila T. Temporal ensembling for semi-supervised learning. arXiv preprint arXiv:1610.02242. 2016 Oct 7.
> > > [2] Xie Q, Dai Z, Hovy E, Luong T, Le Q. Unsupervised data augmentation for consistency training. Advances in neural information processing systems. 2020;33:6256-68.
> > > [3] Sohn K, Berthelot D, Carlini N, Zhang Z, Zhang H, Raffel CA, Cubuk ED, Kurakin A, Li CL. Fixmatch: Simplifying semi-supervised learning with consistency and confidence. Advances in neural information processing systems. 2020;33:596-608.

---

### Official Review · Reviewer_AuP3 · 2024-11-02

**Soundness:** 2
**Presentation:** 1
**Contribution:** 2
**Rating:** 3
**Confidence:** 4

**Summary:**

The authors propose a semi supervised classification based method to perform two sample testing. The problem of two sample testing is as follows: given a finite set of samples from two distributions $P, Q$,  $S_p = (x_1, \dots x_n) \sim P$ and $S_q = (y_1, \dots y_m) \sim Q$, the task is to tell whether $P = Q$ or not. In classifier based two sample testing, a classifier is trained on a split of the data, and the test statistic - the average classification accuracy is estimated on the test split. The null hypothesis, i.e. $P = Q$ is rejected if the test statistic is sufficiently greater than random guessing.

The semi supervised method proposed by the authors instead first learns common features of the samples by training an auto encoder based on reconstruction loss on all of the data in $S_p \cup S_q$. This gives us extracted features $S'_p = (\phi(x_1) , \dots \phi(x_n))$ and $S'_q = (\phi(y_1), \dots \phi(y_m))$. Then the usual classifier based two sample testing is performed on $S'_p, S'_q$.

The authors provide a theorem characterizing a lower bound on test power based on number of unlabeled samples and VC dimension of classifier function class placed on top of featurizer $\phi$, and VC dimension of some compatibility function class that depends on the overall function class (input space to output space).

The authors finally provide some experimental evaluation on some low dimensional synthetic datasets (d = 2 - 10), and two real datasets -  MNIST vs MNIST fake, and Imagenet vs Imagenet fake. On these datasets their method performs better than C2ST.

**Strengths:**

A new method for two sample testing based on a two stage process of extracting features in an unsupervised manner, and then performing C2ST on extracted features instead of original data.

**Weaknesses:**

The paper has several weaknesses.

The overall presentation and organization of the paper is very poor. The actual details of the method appears quite late on page 6. The introduction section is unreasonably long. Figure 2 looks unnecessary. Figure 1 and Figure 3 are repeated. Experiment results are split across multiple sections (Table 1 in introduction, then Table 2, Figure 4 and Figure 5 in Experiments section), whereas they could have been clubbed under Experiments section. Inconsistency in experiments. In table 1 comparison is shown against 5 baselines, whereas in Table 2 it is shown against 4 baselines. Figure 4 looks unnecessary as the results already exist in table 1 and table 2. Proper details on the experimental evaluation are not provided. For instance the details of the model used for feature extractor $\phi$ is not provided. The experimental details of different baselines also not provided clearly. The theorem is quite unclear, notation is missing for instance $\chi(f, x)$ is not defined. No clear understanding of why and when is semi supervised learning better than supervised learning is provided. The proofs in the appendix are quite unclear, for instance proper references to equations is missing.

**Questions:**

1. Definition of $\chi$ in Definition 5.2 is not clear. How is $\chi(f, x)$ defined ? Proof of Theorem 5.3 is also very unclear. What is $\mathcal{C}[2m_1, \mathcal{S}]$, the notation is not defined. $m_1$ is also not defined (although it could be inferred that $m_1$ is number of labeled data).

2. Not clear from the theorem how this gives more power than C2ST ?

3. “Although the differences between two methods are little when N is small, the test power of SSL-C2ST has a huge gap over C2ST when N is large enough” This only seems to be the case for the right most figure ?

---

> ### Author Response · Authors · 2024-11-21
>
> Thanks for reading our paper and for your valuable feedbacks. We will answer your questions and concerns below.
>
> **Weakness 1:**
>
> > The overall presentation and organization of the paper is very poor. The actual details of the method appears quite late on page 6. The introduction section is unreasonably long.
>
> **Response 1:**
>
> We appreciate your feedback regarding the presentation, organization, and the length of the introduction. However, we believe there is a significant **misunderstanding about the primary focus and contribution of our work**, which we would like to clarify here:
>
> Our **main focus / contribution** is not proposing a single new method. Instead, we are **the first to introduce the perspective that the two-sample testing problem (which is a specific case within the field of hypothesis testing) can be framed as a semi-supervised learning (SSL) paradigm**. This novel conceptual contribution addresses the limitations of data-splitting trade-offs inherent in supervised methods and the lack of discriminative power in purely unsupervised methods. This is clearly stated in the ***'Two learning paradigms and their issues'*** section of the Introduction on Page 2 and elaborated further in Section 3 on Page 5.
>
> After developing the intuition that we can combine the field of two-sample testing with the field of semi-supervised learning, since we are the first to propose this novel view, **to validate whether this combination is viable**, we investigate whether existing SSL methods can directly apply to two-sample testing in the Section 4 on Page 5-6. After investigation and experimental verification, existing SSL methods fail due to specific challenges posed by overlapping data distributions in two-sample testing. **Identifying and explaining this limitation** is also a critical part of our contribution.
>
> Based on the findings of our survey and analysis, we propose a viable way to adapt SSL for two-sample testing, which serves as a first step for future research. While this method is simple yet effective, the **conceptual contribution lies in paving the way for future advancements by adapting more advanced SSL methods to the two-sample testing field**.
>
> Our contribution is to establish a **new research direction** that bridges the gap between two previously separate fields, and point out that main limitation of the current supervised paradigm which will discard the information from the unlabeled testing data. We hope this clarification helps to better understand our work and its importance in advancing the field of two-sample testing and hypothesis testing.
>
> **Weakness 2:**
>
> > Figure 2 looks unnecessary. Figure 1 and Figure 3 are repeated.
>
> **Response 2:**
>
> We would like to kindly point out that the importance and necessity of each figure are already explained in the paper. We encourage the reviewer to carefully revisit the respective sections where these figures are discussed. Below, we further clarify their distinct roles to avoid any misunderstanding.
>
> **Figure 2 is critical as it visualizes synthetic data scenarios to mimic the types of data distributions we encounter in two-sample testing experiments.** It highlights the **degree of data overlap**, which directly impacts the viability of state-of-the-art (SOTA) SSL methods for two-sample testing. By clearly showing how the distributions differ and overlap, Figure 2 provides essential context for understanding the challenges our method addresses.
>
> **Figure 1 provides a high-level overview of the existing paradigms in two-sample testing: unsupervised, supervised paradigm.** This figure helps readers conceptually understand the differences between the approaches, framing the motivation for introducing SSL in the context of two-sample testing.
>
> **Figure 3 focuses specifically on our proposed SSL-C2ST pipeline, showing how it integrates the strengths of the original C2ST paradigm while leveraging the information from the unlabeled testing data.** Unlike Figure 1, which provides a broader summary, Figure 3 explains the detailed implementation and workflow of our proposed method.
>
> Each figure has a **unique purpose** and contributes to the overall understanding of our work. We believe their inclusion is essential for a clear and comprehensive presentation of our methodology and experimental setup. We hope this explanation clarifies the necessity and distinct roles of these figures.

---

> > ### Author Response · Authors · 2024-11-21
> >
> > **Weakness 3:**
> >
> > > Experiment results are split across multiple sections (Table 1 in introduction, then Table 2, Figure 4 and Figure 5 in Experiments section), whereas they could have been clubbed under Experiments section. Inconsistency in experiments. In table 1 comparison is shown against 5 baselines, whereas in Table 2 it is shown against 4 baselines.
> >
> > **Response 3:**
> >
> > We believe that if **Response 1**, which clarifies the main contribution of our work, is understood, then the purpose of Table 1 becomes clear. Table 1 is not included to demonstrate the performance of our proposed method **but rather serves as the motivation experiments to validate the intuition that can we directly apply SSL techniques on the two-sample testing methods**. Therefore, it is more appropriate to position Table 1 outside the experimental section.
> >
> > On the other hand, **Table 2 focuses on comparing the performance of our proposed methods against other SOTA two-sample testing methods,** presenting the main empirical results. Thus, there is no inconsistency in their scope or use.
> >
> > **Weakness 4:**
> >
> > > Figure 4 looks unnecessary as the results already exist in table 1 and table 2.
> >
> > **Response 4:**
> >
> > We would like to clarify that **Figure 4 is essential and serves a distinct purpose, separate from Table 1 and Table 2.** First and foremost, **Figure 4 is not related to Table 1.** Table 1 focuses on the motivation experiments to demonstrate the viability of our conceptual contribution, while Figure 4 highlights an ablation study specific to our proposed method. We kindly encourage the reviewer to carefully revisit the paper, as this distinction is clearly explained.
> >
> > Unlike Table 1, which evaluates the applicability of existing SSL methods to two-sample testing, and Table 2, which presents the main empirical results comparing our proposed method with other SOTA two-sample testing methods, **Figure 4 focuses on the ablation study.** This figure clearly displays the significant improvements of an additional step to the original C2ST method (as part of our SSL-C2ST pipeline).
> >
> > **Figure 4 evaluates our method on two widely used benchmarks in two-sample testing, HDGM-Hard (with different dimensions) and MNIST vs Fake-MNIST datasets.** These benchmarks are critical for demonstrating the incremental value of our approach and explaining why the proposed SSL-C2ST pipeline is effective. **Removing Figure 4 would result in the loss of critical insights from the ablation study,** which are necessary to fully support the design decisions and effectiveness of our method.
> >
> > We hope this explanation clarifies the unique and important role of Figure 4 and addresses the misunderstanding regarding its connection to Table 1.

---

> > > ### Author Response · Authors · 2024-11-21
> > >
> > > **Weakness 5:**
> > >
> > > >  Proper details on the experimental evaluation are not provided.  For instance the details of the model used for feature extractor ϕ is not provided. The experimental details of different baselines also not provided clearly.
> > >
> > > **Response 5:**
> > >
> > > We would like to clarify that the details regarding the experimental evaluation are provided in the paper and its appendix.
> > >
> > > As shown in **Figure 3** on Page 7, which provides an overview of our proposed paradigm, we explicitly state that the encoder ϕ from the AE-based representation learning shares **exactly the same structure as the feature extractor in the classifier model of the original C2ST**. Since the reference to the related work about C2ST is clearly cited, we believe it is unnecessary to repeat the same structure already described in the referenced work. However, if the reviewer still believes this information would benefit readers who are less familiar with the field of two-sample testing, we are happy to add the structure in the appendix.
> > >
> > > We have also clearly stated in the paper that the **implementation details and parameter settings for all baselines** can be found in **Appendix C.2 and C.3 on Pages 16–17.** These sections provide a thorough description of how we implemented the methods and the parameter configurations we used for our comparisons.
> > >
> > > We hope this explanation clarifies that the relevant details are indeed included in the paper. Should the reviewer require further clarification, we would be happy to make the references to these details in the appendix more prominent in the revised version.
> > >
> > > **Weakness 6:**
> > >
> > > > The theorem is quite unclear, notation is missing for instance χ(f,x) is not defined. No clear understanding of why and when is semi supervised learning better than supervised learning is provided. The proofs in the appendix are quite unclear, for instance proper references to equations is missing.
> > >
> > > **Response 6:**
> > >
> > > Thank you for your comments regarding the clarity of the theorem and proofs. Below, we address each concern in detail:
> > >
> > > χ(f,x) is clearly **defined in Definition 5.2 on Page 8** as the compatibility which measures the distance between x and its reconstructed x' by feature extractor and decoder.
> > >
> > > As stated in the Theorem 5.3, with a **high probability of 1−δ**, when the number of unlabeled $m_u$ and labeled $m_l$ samples is sufficiently large, compared to the purely supervised learning paradigm, as the unlabeled data in training is increasing, the semi-supervised learning paradigm achieves a decreasing upper bound of the empirical $ \epsilon$, which increases the lower bound of test power. As under the alternative hypothesis $H_1$, when $m_l$ increases, the test power of two-sample testing will always converge to 1, increasing the lower bound of test power without increasing $m_l$ is the improvement than original C2ST.
> > >
> > > Regarding the proofs in the appendix, we are unsure which specific references to equations are missing or unclear. If the reviewer could specify the particular parts that seem unclear, we would be happy to revise and ensure all necessary references are included in the proofs for better readability and understanding.

---

> ### Author Response · Authors · 2024-11-21
>
> **Question 1:**
>
> > Definition of χ in Definition 5.2 is not clear. How is χ(f,x) defined ? Proof of Theorem 5.3 is also very unclear. What is C[2m1,S], the notation is not defined. m1 is also not defined (although it could be inferred that m1 is number of labeled data).
>
> **Response 1:**
>
> Firstly, **χ** represents the compatibility, which is clearly defined in **Definition 5.2 on Page 8** and directly refers to the work by [1] under our transductive learning settings. We encourage the reviewer to revisit this definition for further clarification.
>
> The notation **C[2$m_l$, S]** is defined around **line 430** as the **expected split number for 2$m_l$ points drawn from S using functions in C**. Additionally, it is **m_l (the labeled sample size)**, not **m_1** as inferred. This is explicitly defined on **line 428** of the paper.
>
> We hope this clarification resolves the concerns regarding the definitions and proof notations. Please let us know if further elaboration is required.
>
> [1] Maria-Florina Balcan and Avrim Blum. A discriminative model for semi-supervised learning. Journal of the ACM, 57(3):19:1–19:46, 2010
>
> **Question 2:**
>
> > Not clear from the theorem how this gives more power than C2ST ?
>
> **Response 2:**
>
> Please refer to the Response 6 to Weakness 6.
>
> **Question 3:**
>
> > “Although the differences between two methods are little when N is small, the test power of SSL-C2ST has a huge gap over C2ST when N is large enough” This only seems to be the case for the right most figure ?
>
> **Response 3:**
>
> As you can see clearly from the **y-axis of Figure 4**, the gap appears larger in the right-most subfigure because **HDGM-Hard with dimension 10 is a more challenging dataset**, making it harder for C2ST to converge to 1 with a relatively small number of input samples compared to SSL-C2ST.
>
> For the other two relatively easier datasets shown in the other subfigures, **the huge gap lies in the number of samples required for convergence.** While SSL-C2ST converges to 1 with relatively fewer input samples, C2ST requires a significantly larger amount of data to achieve the same convergence. This difference reflects the **performance gap** between the two methods.
>
> In two-sample testing, under the alternative hypothesis, as the number of samples increases, the test power of an advanced model will eventually converge to 1. **If a model requires significantly fewer samples to reach this convergence, it is considered to be outperforming.**
>
> We hope this explanation provides additional clarity regarding the observed gaps in Figure 4.
>
> **We would be delighted to engage in further discussions to ensure your concerns are fully addressed.**

---

> > ### Comment · Reviewer_AuP3 · 2024-11-24
> > **Response to rebuttal by authors**
> >
> > **"the first to introduce the perspective that the two-sample testing problem (which is a specific case within the field of hypothesis testing) can be framed as a semi-supervised learning (SSL) paradigm"**
> >
> > I don't see novelty in introducing the perspective that SSL can be used in two-sample testing. The metric of judgement in my view would remain in the method proposed, which I see being introduced quite late in the paper.
> >
> > The theorems are still not clear, the authors say "χ(f,x) is clearly defined in Definition 5.2 on Page 8", however I still don't see a clear definition of χ(f,x)".
> >
> > I recommend authors to rewrite the paper mentioning their key contributions and results in a succinct manner without building a lot of motivation.
> >
> > Given the lack of proper presentation and clarity in theorems, I would like to keep my current score.

---

> ### Author Response · Authors · 2024-11-24
>
> Dear Reviewer AuP3,
>
> Thanks for your reply. For compatibility, it is indeed defined in Definition 5.2 on Page 8. We are not sure if you indicated to give some examples regarding this function. If so, we are happy to give it to you. it is just a function, and we defined its input and output spaces already.
>
> As for the novelty, we **respectfully disagree with you**, and your judgment is subjective. Our main contribution lies in that **we find a new perspective to study the two-sample test**, where the two-sample test and corresponding techniques are one of the most important tools used in the machine learning field. Thus, **proposing a new research direction for the two-sample test, an important problem, is novel and significant.** This will light up more research studies in this field, by following our novel perspective.
>
> Our novelty is fully recognized by three reviewers:
>
> 1) Reviewer Y9aR: "The paper is **novel in applying a semi-supervised learning framework to the two-sample testing problem**, a **first** in this area. The proposed framework is simple yet effective, as demonstrated by the experimental results."
>
> 2) Reviewer 6c7X: "The authors clearly highlight the issues with the current methodologies, particularly when viewed from the bucket of supervised and unsupervised methods. **The proposed approach cleverly circumvents this problem for both buckets.** The applicability to different scenarios seems to be the biggest plus point of using SSL based testing methods."
>
> 3) Reviewer 47NZ: "This paper proposes **a novel perspective** to consider non-parametric two-sample testing as a semi-supervised learning (SSL) problem."
>
> We hope to further discuss with you and other reviewers the position of our paper in the field of two-sample testing. **We believe this is the major divergence between you and us. Thus, it is important to find a consensus toward the novelty judgment.**
>
> Best regards,
>
> Senior Author of Submission 6635

---

> > ### Author Response · Authors · 2024-11-26
> >
> > Dear Reviewer AuP3,
> >
> > Thank you again for your time and effort in reviewing our paper.
> >
> > We have carefully addressed all your comments and prepared detailed responses. Could you kindly review them at your earliest convenience?
> >
> > As the discussion period is coming to a close, we wanted to check if there are any remaining concerns preventing you from adjusting your score. If there is any additional clarification we can provide, please do not hesitate to let us know, and we will address it promptly.
> >
> > Best regards,
> >
> > Authors of Submission 6635

---

> > > ### Author Response · Authors · 2024-11-29
> > >
> > > Dear Reviewer AuP3,
> > >
> > > Thank you for your valuable feedback and constructive comments on our submission.
> > >
> > > Since we have already solve all the concerns that has been proposed, we would like to ask if you have any additional concerns or questions about the paper that we can address. If there are specific points that remain unclear, please feel free to point them out—we are more than willing to discuss and provide further explanations.
> > >
> > > If all your concerns have been addressed satisfactorily, we would sincerely appreciate it if you could consider revisiting your scores and generously increasing them, if appropriate, to reflect the clarified contributions of the paper.
> > >
> > > Thank you once again for your thoughtful review and for helping us improve our work.
> > >
> > > Best regards,
> > >
> > > Authors of Submission 6635

---

> > > > ### Author Response · Authors · 2024-11-30
> > > >
> > > > Dear Reviewer AuP3,
> > > >
> > > > We hope this message finds you well. As the closing date for the review process is approaching, we wanted to follow up to ensure that all of your concerns regarding our paper have been addressed. If there are any remaining questions or issues you would like us to clarify, we would be more than happy to discuss them further.
> > > >
> > > > If all your concerns have been resolved, we kindly ask if you could consider revisiting your scores and updating them to reflect the clarified contributions of our work. Your constructive feedback has been invaluable, and we sincerely appreciate the time and effort you have devoted to reviewing our paper.
> > > >
> > > > Thank you once again for your thoughtful evaluations, and we look forward to hearing from you.
> > > >
> > > > Best regards,
> > > >
> > > > Authors of Submission 6635

---

> > > > > ### Author Response · Authors · 2024-12-01
> > > > >
> > > > > Dear Reviewer AuP3,
> > > > >
> > > > > We hope this message finds you well. As the closing date for the review process is approaching, we wanted to follow up to ensure that all of your concerns regarding our paper have been addressed. If there are any remaining questions or issues you would like us to clarify, we would be more than happy to discuss them further.
> > > > >
> > > > > If all your concerns have been resolved, we kindly ask if you could consider revisiting your scores and updating them to reflect the clarified contributions of our work. Your constructive feedback has been invaluable, and we sincerely appreciate the time and effort you have devoted to reviewing our paper.
> > > > >
> > > > > Thank you once again for your thoughtful evaluations, and we look forward to hearing from you.
> > > > >
> > > > > Best regards,
> > > > >
> > > > > Authors of Submission 6635

---

> > > > > > ### Author Response · Authors · 2024-12-02
> > > > > >
> > > > > > Dear Reviewer AuP3,
> > > > > >
> > > > > > We hope this message finds you well. As the closing date for the review process is approaching, we wanted to follow up to ensure that all of your concerns regarding our paper have been addressed. If there are any remaining questions or issues you would like us to clarify, we would be more than happy to discuss them further.
> > > > > >
> > > > > > If all your concerns have been resolved, we kindly ask if you could consider revisiting your scores and updating them to reflect the clarified contributions of our work. Your constructive feedback has been invaluable, and we sincerely appreciate the time and effort you have devoted to reviewing our paper.
> > > > > >
> > > > > > Thank you once again for your thoughtful evaluations, and we look forward to hearing from you.
> > > > > >
> > > > > > Best regards,
> > > > > >
> > > > > > Authors of Submission 6635

---

> > > > > > > ### Comment · Reviewer_AuP3 · 2024-12-02
> > > > > > >
> > > > > > > Thank you for your response. I have considered the arguments and would like to keep the current rating.

---

### Official Review · Reviewer_6c7X · 2024-11-04

**Soundness:** 3
**Presentation:** 3
**Contribution:** 3
**Rating:** 6
**Confidence:** 2

**Summary:**

The paper proposes a new approach towards non-parametric two-sample testing, by first learning inherent representations using both labeled and unlabeled data, and then fine-tuning these representations by additionally learning discriminative representations using labeled data. The effectiveness of this methodology over traditional classifier two-sample testing is demonstrated via experiments and theory. The intuition behind additional utilization of unlabeled data seems solid, and the results are commendable.

P.S. My expertise in two-sample testing is limited, thus I am judging the paper mostly from the angle of how effectively the representations are learnt, thus unsure about the contributions and novelty in the given sub-field but curious to hear from other reviewers.

**Strengths:**

- [S1] The authors clearly highlight the issues with the current methodologies, particularly when viewed from the bucket of supervised and unsupervised methods. The proposed approach cleverly circumvents this problem for both buckets. The applicability to different scenarios seems to be the biggest plus point of using SSL based testing methods.
- [S2] The authors also negate the initial thoughts one might have on reading the setup and motivation by discussing the failure modes for SOTA SSL methods in this two-sample testing scenario.
- [S3] The proofs add a lot of value to the paper, and along with the provided code, make the submission a quite complete pitch.

**Weaknesses:**

- [W1] I think some reorganization might help the paper read better, for example including the Algorithm 1 in the main paper perhaps at the cost of some preliminaries or shortening the introduction could help understand the proposed method better. Right now, while the motivation and setup comes across very clearly, the details of the approach are somehow missed, and the key points of why the method works in practice, while written in words, are not evident algorithm wise.
- [W2] On a similar note, I’m slightly unsure where the exact improvements are coming from, is it possible to look at some ablations where one of the steps was modified, say the featurizer for learning IRs was an suboptimal one, how would the learning DRs be affected? It would be helpful to understand the utility of each part of the sequential method proposed by the authors.

**Questions:**

- [Q1] For Table 1, I understand there is no standard deviation across one sampling of two groups, but would there not be several runs of such samplings that could be aggregated against and then the standard deviation be expressed? While the result of each trial is 0 or 1, would it not be more insightful to have several trials (say 5) and report the results across them in total?

---

> ### Author Response · Authors · 2024-11-21
>
> Thanks for reading our paper and for your valuable feedbacks. We will answer your questions and concerns below.
>
> **Weakness 1:**
>
> > I think some reorganization might help the paper read better, for example including the Algorithm 1 in the main paper perhaps at the cost of some preliminaries or shortening the introduction could help understand the proposed method better. Right now, while the motivation and setup comes across very clearly, the details of the approach are somehow missed, and the key points of why the method works in practice, while written in words, are not evident algorithm wise.
>
> **Response 1:**
>
> We appreciate the reviewer’s suggestion regarding the paper’s organization, and we agree that **reorganizing Section 3** which has some repeated contents to the introduction could improve the paper's readability. Specifically, we can shorten the main paper by concisely restating the key motivation and emphasizing the two research questions addressed in this paper.
>
> The freed-up space can then be used to **include Algorithm 1 in the main paper,** which will provide clearer support for the details of the approach and make the key points of why the method works in practice more evident. We believe these adjustments will make the paper more organized and easier to follow while addressing the concern about missing algorithmic details.
>
> **Weakness 2:**
>
> > On a similar note, I’m slightly unsure where the exact improvements are coming from, is it possible to look at some ablations where one of the steps was modified, say the featurizer for learning IRs was an suboptimal one, how would the learning DRs be affected? It would be helpful to understand the utility of each part of the sequential method proposed by the authors.
>
> **Response 2:**
>
> The improvement in performance comes from the fact that the feature extractor learned from the autoencoder will be effective when our data well fits the manifold assumption, that is **high dimensional data lies approximately on a lower-dimensional manifold**. Moreover, effective and accurate lower-dimensional data representation will improve the capability of detecting discrepancy in the MMD testing. As found in [1], **the faster the actual dimension decreases relative to the sample size, the higher-order moment discrepancy the MMD test are capable to detect**, so MMD-based SSL-C2ST (SSL-C2ST-M) can be such outperforming than the other SOTA two-sample testing methods across all the benchmarks.
>
> For the ablation study, as shown in **Figure 3**, compared to the original C2ST pipeline, our improvement lies in pre-training the featurizer on the entire labeled and unlabeled dataset, which will learn a better IRs than that of C2ST. This step allows the featurizer to become more general and capable of mapping the data onto an effective low-dimensional manifold, which facilitates the following learning discriminative representations (DRs). The utility of this additional step is also demonstrated empirically in the **Figure 4** in the experiments section, where we show how incorporating this pre-training step leads to improved test power than that of the original C2ST.
>
> We hope this explanation clarifies the utility of each part of the sequential method and the source of the observed improvements.
>
> [1] Yan J, Zhang X. Kernel two-sample tests in high dimensions: interplay between moment discrepancy and dimension-and-sample orders. Biometrika. 2023 Jun 1;110(2):411-30.

---

> ### Author Response · Authors · 2024-11-21
>
> **Question 1:**
>
> > For Table 1, I understand there is no standard deviation across one sampling of two groups, but would there not be several runs of such samplings that could be aggregated against and then the standard deviation be expressed? While the result of each trial is 0 or 1, would it not be more insightful to have several trials (say 5) and report the results across them in total?
>
> **Response 1:**
>
> We appreciate the reviewer’s question regarding the reporting of standard deviation in Table 1. To clarify, we **indeed conducted each experiment 100 times, or to say sampling 100 trials from the two distributions.** However, in each sampling, the result is either 0 or 1, meaning that the standard deviation becomes quite large and **less meaningful** when the input sample size is small. Moreover, if you suggest to perform the permutation testing several times, we also indeed perform permutation testing 100 times for each sampling, but since our problem setting is transductive learning, so the result is still either 0 or 1, or very close to these two values.
>
> To address this, we only use **test power** to represent the results. Test power reflects the **estimated probability of successfully discriminating (i.e., returning 1)** between the two samples across the 100 trials. This provides a clearer and more interpretable metric for our experiments.
>
> We hope this explanation clarifies our choice of representation in Table 1.
>
> **We would be delighted to engage in further discussions to ensure your concerns are fully addressed.**

---

> > ### Comment · Reviewer_6c7X · 2024-11-22
> > **Reply to the Rebuttal**
> >
> > I appreciate the detailed rebuttal.
> >
> > - [W1] Noted.
> > - [W2] Ah I see, I am still slightly on the fence of the innovation in this methodology and reading more on the background literature to make an informed choice.
> > - [Q1] That seems to be a reasonable justification
> >
> > I am hesitant to make a stronger recommendation as of now, given my limited exposure to this area. I am going through the other reviews and rebuttals, and looking forward to the discussion period.

---

> > > ### Author Response · Authors · 2024-11-22
> > >
> > > Dear Reviewer 6c7X,
> > >
> > > Many thanks for your response! It is glad to know your concerns are almost addressed.
> > >
> > > As for the innovation of our paper, **the main novelty of our paper is to revisit the two-sample testing as a semi-supervised learning (SSL) problem**, which is a **conceptual-level contribution** to the field of two-sample testing. As the first investigation toward this research direction, we did a lot of explorations and found that direct use of existing SSL methods did not work well in two-sample testing scenarios. The main reason is that the two samples (i.e., two classes in the classification view) largely overlapped in the testing scenarios, such that the assumptions of many SSL methods cannot be satisfied. In the end, we find a suitable SSL way to use the unlabelled data: first learn a good representation (i.e., IRs) via unlabelled data, then learn DRs.
> > >
> > > **Why our method works well? How does the unlabelled data help?** From an SSL view, if we learn good IRs for data, we actually handle data with relatively lower dimensions compared to the original data, making the problem easier to address. From a two-sample testing view, as found in [1], the faster the actual dimension decreases relative to the sample size, the higher-order moment discrepancy the MMD test are capable to detect (i.e., MMD test can work better when meeting lower-dimension data).
> > >
> > > Hope the above explanations can help you further re-evaluate our paper's contribution ^^. Your recommendation to accept (instead of a borderline decision) will be much appreciated.
> > >
> > > [1] Yan J, Zhang X. Kernel two-sample tests in high dimensions: interplay between moment discrepancy and dimension-and-sample orders. Biometrika. 2023 Jun 1;110(2):411-30.
> > >
> > > Best regards,
> > >
> > > The authors of Submission6635

---

> > > > ### Author Response · Authors · 2024-11-26
> > > >
> > > > Dear Reviewer 6c7X,
> > > >
> > > > Thank you again for your time and effort in reviewing our paper.
> > > >
> > > > We have carefully addressed all your comments and prepared detailed responses. Could you kindly review them at your earliest convenience?
> > > >
> > > > As the discussion period is coming to a close, we wanted to check if there are any remaining concerns preventing you from adjusting your score. If there is any additional clarification we can provide, please do not hesitate to let us know, and we will address it promptly.
> > > >
> > > > Best regards,
> > > >
> > > > Authors of Submission 6635

---

> > > > > ### Author Response · Authors · 2024-11-29
> > > > >
> > > > > Dear Reviewer 6c7X,
> > > > >
> > > > > Thank you for your valuable feedback and constructive comments on our submission.
> > > > >
> > > > > Since we have already solve all the concerns that has been proposed, we would like to ask if you have any additional concerns or questions about the paper that we can address. If there are specific points that remain unclear, please feel free to point them out—we are more than willing to discuss and provide further explanations.
> > > > >
> > > > > If all your concerns have been addressed satisfactorily, we would sincerely appreciate it if you could consider revisiting your scores and generously increasing them, if appropriate, to reflect the clarified contributions of the paper.
> > > > >
> > > > > Thank you once again for your thoughtful review and for helping us improve our work.
> > > > >
> > > > > Best regards,
> > > > >
> > > > > Authors of Submission 6635

---

> > > > > > ### Author Response · Authors · 2024-11-30
> > > > > >
> > > > > > Dear Reviewer 6c7X,
> > > > > >
> > > > > > We hope this message finds you well. As the closing date for the review process is approaching, we wanted to follow up to ensure that all of your concerns regarding our paper have been addressed. If there are any remaining questions or issues you would like us to clarify, we would be more than happy to discuss them further.
> > > > > >
> > > > > > If all your concerns have been resolved, we kindly ask if you could consider revisiting your scores and updating them to reflect the clarified contributions of our work. Your constructive feedback has been invaluable, and we sincerely appreciate the time and effort you have devoted to reviewing our paper.
> > > > > >
> > > > > > Thank you once again for your thoughtful evaluations, and we look forward to hearing from you.
> > > > > >
> > > > > > Best regards,
> > > > > >
> > > > > > Authors of Submission 6635

---

> > > > > > > ### Author Response · Authors · 2024-12-01
> > > > > > >
> > > > > > > Dear Reviewer 6c7X,
> > > > > > >
> > > > > > > We hope this message finds you well. As the closing date for the review process is approaching, we wanted to follow up to ensure that all of your concerns regarding our paper have been addressed. If there are any remaining questions or issues you would like us to clarify, we would be more than happy to discuss them further.
> > > > > > >
> > > > > > > If all your concerns have been resolved, we kindly ask if you could consider revisiting your scores and updating them to reflect the clarified contributions of our work. Your constructive feedback has been invaluable, and we sincerely appreciate the time and effort you have devoted to reviewing our paper.
> > > > > > >
> > > > > > > Thank you once again for your thoughtful evaluations, and we look forward to hearing from you.
> > > > > > >
> > > > > > > Best regards,
> > > > > > >
> > > > > > > Authors of Submission 6635

---

> > > > > > > > ### Comment · Reviewer_6c7X · 2024-12-02
> > > > > > > >
> > > > > > > > I appreciate the authors' effort to reply and engage. While I do understand the clarifications authors' have provided, I still don't cannot commend a clear acceptance due to my limited background in certain related areas.
> > > > > > > >
> > > > > > > > I do feel that the concerns raised by fellow reviewers (to name one: novelty over fine-tuning based techniques, assumptions for high dimensional data) are legitimate and unfortunately cannot completely overlook them.
> > > > > > > >
> > > > > > > > I do feel the paper is still a step in the positive direction, but with certain flaws, thus will keep the borderline accept rating for now.

---

### Official Review · Reviewer_Y9aR · 2024-11-04

**Soundness:** 3
**Presentation:** 4
**Contribution:** 2
**Rating:** 6
**Confidence:** 3

**Summary:**

The authors introduce SSL-based classifier two-sample test framework that addresses limitations of discriminative representations (DRs) and inherent representations (IRs). DRs reduce the data points available for testing phase whereas IRs do not provide discriminative cues.
Both IRs and DRs need some supervision to learn good representation (DRs) or good candidate kernels (IRs). For this, the authors propose a semi-supervised learning (SSL) framework, which utilizes features learned in an unsupervised manner to fit additional classifier head to maximize the test power.

**Strengths:**

- Writing: The task of two-sample testing is clearly defined, with concise notations and a clear outline of the limitations of existing supervised and unsupervised approaches.
- Novelty: The paper is novel in applying a semi-supervised learning framework to the two-sample testing problem, a first in this area. The proposed framework is simple yet effective, as demonstrated by the experimental results.
- Proof of concept: Section 4, Figure 2, and Table 1 illustrate why a straightforward application of existing SSL methods may fail in certain cases, such as the HDGM-medium and HDGM-hard tasks. This clarifies the motivation for the proposed SSL-C2ST approach.
- Theoretical justification: The authors provides why more unlabeled data points can help improving test power.
- Empirical support: The experiment results on several datasets show the superiority of the proposed method over existing state-of-the-art two-sample testing methods.

**Weaknesses:**

* Writing: The limitations of DRs and IRs, along with the motivation for the proposed method, are reiterated throughout Sections 1 and 3 without providing additional insights. Although recognizing these problems is key to the paper’s contributions, a single, well-placed explanation could make the writing more concise.
* Novelty: The proposed SSL-C2ST pipeline is straightforward and highly effective, but it closely resembles standard fine-tuning practices, where only the last few layers are updated after learning self-supervised features — a common approach in neural network training.
* Ambiguity: It is not very clear if obtaining features in an unsupervised manner is classified as SSL. Related to this, what do the authors think the underlying assumption of auto-encoder features is for SSL tasks if none of the three assumptions apply for this?

**Questions:**

1. Could you elaborate the equivalence between$max_\epsilon \frac{ (1/2- \epsilon) }{ \sqrt{\epsilon - \epsilon^2} }$ and $min_\epsilon \frac{\epsilon} {\sqrt{\epsilon - \epsilon^2} }$? It is not very obvious to me because of the denaminator.
2. In Theorem 5.3, I understand as $m_u$ increases, the third term in Eq.(8) decreases but how does that affect $\epsilon(P, Q; f’^*)$? I wonder 1. $\epsilon(P, Q; f’^*)$ does not depend on $m_u$, 2. If so, when $\epsilon(P, Q; f’^*)$ is large, is the upper bound effective?
3. Auto-encoder is not the only way to obtain unsupervised-learning features. In fact, contrastive learning is perhaps more popular and effective. Is there any particular reason why the authors restricted unsupervised-learning features to be auto-encoder-based features?

---

> ### Author Response · Authors · 2024-11-21
>
> Thanks for reading our paper and for your valuable feedbacks. We will answer your questions and concerns below.
>
> **Weakness 1:**
>
> > Writing: The limitations of DRs and IRs, along with the motivation for the proposed method, are reiterated throughout Sections 1 and 3 without providing additional insights. Although recognizing these problems is key to the paper’s contributions, a single, well-placed explanation could make the writing more concise.
>
> **Response 1:**
>
> We appreciate the reviewer’s feedback regarding the writing and organization of Sections 1 and 3. **We agree that emphasizing the limitations of supervised and unsupervised paradigms is key to motivating our work,** but we see an opportunity to improve conciseness.
>
> To address this, we will reorganize **Section 3** to provide a concise summary of these limitations and motivations, followed by the pointing out the **two research questions** we aim to address. This adjustment will make the paper more organized and easier to follow.
>
> **Weakness 2:**
>
> > Novelty: The proposed SSL-C2ST pipeline is straightforward and highly effective, but it closely resembles standard fine-tuning practices, where only the last few layers are updated after learning self-supervised features — a common approach in neural network training.
>
> **Response 2:**
>
> We appreciate the reviewer’s observation regarding the simplicity of the SSL-C2ST pipeline. As we restate in the paper, the main goal of our work is to **pave the way for future research** on framing semi-supervised learning techniques within the two-sample testing field, e.g. testing the viability of directly applying SOTA SSL techniques on the two-sample testing methods, proposing a simple but viable method.
>
> Our primary contribution is to demonstrate that this perspective is **worthy of further exploration** and that even a simple method can effectively improve performance in this context. We also highlight that **methods with strict assumptions often struggle in two-sample testing scenarios,** making this an important area for developing more advanced, adapted SSL techniques.
>
> **Weakness 3:**
>
> > Ambiguity: It is not very clear if obtaining features in an unsupervised manner is classified as SSL. Related to this, what do the authors think the underlying assumption of auto-encoder features is for SSL tasks if none of the three assumptions apply for this?
>
> **Response 3:**
>
> We would like to clarify the assumptions regarding two-sample testing data and their relevance to our method. **As discussed in the paragraph "Testing data might not satisfy the assumptions made by many SSL methods" on Page 6, the data violates the smoothness and cluster assumptions but satisfies the manifold assumption.**
>
> The choice of autoencoder-based representation learning is motivated by two main considerations:
>
> 1. **Alignment with C2ST architecture**: In the original C2ST model, a feature extractor is included. By incorporating an autoencoder, we leverage the additional unlabeled testing data to enhance the feature extractor’s capability.
>
> 2. **Relaxed assumptions**: Unlike many other SSL techniques, which depend on fulfilling multiple assumptions (e.g., smoothness, cluster, and manifold), the effectiveness of an autoencoder primarily relies on the manifold assumption, which is satisfied in two-sample testing scenarios.
>
> Additionally, we would like to restate the **original definition of SSL setting**, as defined in [1]: *SSL can leverage the information P(x) from unlabeled data to help the inference of P(y|x).* If the unlabeled data degrades prediction accuracy by misguiding the inference (e.g., due to violating the assumptions of SSL techniques), then that cannot be classified as effective SSL method. This definition is critical because **it is aligned with our SSL setting, which we utilize the information of unlabeled data to improve the prediction accuracy.**
>
> We acknowledge that the original definition of SSL might not have been clearly reiterated in the paper. We will ensure to emphasize it and restate the reasons for using AE-based representation learning in the revised version.
>
> [1] Olivier Chapelle, Bernhard Schölkopf, and Alexander Zien (eds.). *Semi-Supervised Learning.* The MIT Press, 2006. ISBN 9780262033589.

---

> ### Author Response · Authors · 2024-11-21
>
> **Question 1:**
>
> > Could you elaborate the equivalence between $\max_{\epsilon} (\frac{1/2 - \epsilon}{\sqrt{\epsilon - \epsilon ^2}})$ and $\min_{\epsilon} (\frac{\epsilon}{\sqrt{\epsilon - \epsilon ^2}})$? It is not very obvious to me because of the denominator.
>
> **Response 1:**
>
> Sorry for a direct transformation of the equation, and we will provide the detailed explanation of the equivalence.
>
> If we define $f(\epsilon) = \frac{1/2 - \epsilon}{\sqrt{\epsilon - \epsilon ^2}}$,  $g(\epsilon) = \frac{\epsilon}{\sqrt{\epsilon - \epsilon ^2}}$ and $D(\epsilon) = \sqrt{\epsilon - \epsilon^2}$, where $\epsilon \in (0, \frac{1}{2})$.
>
> The equation $\max_{\epsilon} f(\epsilon) = \max_{\epsilon} \left(\frac{1/2}{D(\epsilon)} - g(\epsilon)\right)$ holds. It is clear to find that $f(\epsilon)$ and $\frac{1/2}{D(\epsilon)}$ are monotonically decreasing over the domain of $\epsilon$. Thus, only if $g(\epsilon)$ is monotonically increasing over the domain of $\epsilon$, the equation $\max_{\epsilon} \left(\frac{1/2}{D(\epsilon)} - g(\epsilon)\right) = \max_{\epsilon} \left(\frac{1/2}{D(\epsilon)}\right) - \min_{\epsilon} \left(g(\epsilon)\right)$ holds. We can understand the ambiguity that why the numerator and denominator of $g(\epsilon)$ are both increasing functions, but we declare the monotonic increasing behavior of entire $g(\epsilon)$. The reason why we say that is over the domain of $\epsilon$, we can find that the increasing rate of numerator is larger than that of denominator. Moreover, we can provide the proof of this phenomenon by simply calculating the derivative of $g(\epsilon)$ w.r.t $\epsilon$. Firstly, let us calculate the derivative of $D(\epsilon) = \left(\epsilon - \epsilon^2\right)^{1/2}$ w.r.t $\epsilon$,
> $$
> \begin{align*}
>     D'(\epsilon) &= \frac{1}{2(\epsilon - \epsilon^2)^{1/2}} \cdot (1 - \epsilon + (-\epsilon)) \\\\
>     &= \frac{1 - 2\epsilon}{2D(\epsilon)},
> \end{align*}
> $$
>
> then, we take the derivative of $g(\epsilon) = \frac{\epsilon}{D(\epsilon)}$ w.r.t $\epsilon$,
> $$
> \begin{align*}
> g'(\epsilon) &= \frac{1 \cdot D(\epsilon) - \epsilon \cdot D'(\epsilon)}{D(\epsilon)^2} = \frac{1}{D(\epsilon)^2} \cdot \frac{2D(\epsilon)^2 - \epsilon(1-2\epsilon)}{2D(\epsilon)} \\\\
> &= \frac{1}{\epsilon(1-\epsilon)} \cdot \frac{2(\epsilon - \epsilon^2) - \epsilon(1-2\epsilon)}{2D(\epsilon)} \\\\
> &= \frac{\epsilon}{\epsilon(1-\epsilon) \cdot 2\sqrt{\epsilon(1-\epsilon)}} = \frac{1}{2(1-\epsilon)\sqrt{\epsilon(1-\epsilon)}}.
> \end{align*}
> $$
> We can find that over the domain of $\epsilon \in (0, \frac{1}{2})$, $g'(\epsilon) > 0$, which concludes the proof. The reason why we want to derive the objective to be equivalent to $\min_{\epsilon}\frac{\epsilon}{\sqrt{\epsilon - \epsilon ^2}}$ is we can simplify it to $\min_{\epsilon}\sqrt{\frac{\epsilon}{(1-\epsilon)}} = \min_{\epsilon}\frac{\epsilon}{(1-\epsilon)}$, where $\epsilon \in (0, \frac{1}{2})$. In that way, it is quite straightforward to understand how minimizing $\epsilon$ can help to improve test power.
>
> In order to eliminate ambiguity, we will also add this proof in the revised paper.

---

> > ### Author Response · Authors · 2024-11-21
> >
> > **Question 2:**
> >
> > > In Theorem 5.3, I understand as mu increases, the third term in Eq.(8) decreases but how does that affect ϵ(P,Q;f′∗)? I wonder 1. ϵ(P,Q;f′∗) does not depend on mu, 2. If so, when ϵ(P,Q;f′∗) is large, is the upper bound effective?
> >
> > **Response 2:**
> >
> > We appreciate the reviewer’s question regarding the effectiveness of the Theorem 5.3. To clarify, $\epsilon(P, Q; f'^*)$ is a fixed value, representing the inability of the unknown best model $f'^*$ on the entire distributions P and Q. It does not depend on $m_u$.
> >
> > Furthermore, we should acknowledge that a minor adjustment is needed in the inequality in order to fix this misunderstanding that will the $\epsilon(P, Q; f'^*)$ large enough to make the upper bound ineffective. The inequality is derived from $2\hat{\epsilon}(S_P, S_Q; f') \leq 2\epsilon(P, Q; f'^*) + \Delta_{m_u, m_l} + \sqrt{\ln(\frac{4}{\delta})/2m_u}$. However, we simplify it by $\hat{\epsilon}(S_P, S_Q; f') \leq 2\hat{\epsilon}(S_P, S_Q; f')$ and derive the Eq.(8), which will make the upper bound meaningless if the $2\epsilon(P, Q; f'^*) > \hat{\epsilon}(S_P, S_Q; f')$. We will revise it into the correct form that ensures the optimal value in the upper bound is always smaller than or equal to the left part of the inequality.

---

> ### Author Response · Authors · 2024-11-21
>
> **Question 3:**
>
> > Auto-encoder is not the only way to obtain unsupervised-learning features. In fact, contrastive learning is perhaps more popular and effective. Is there any particular reason why the authors restricted unsupervised-learning features to be auto-encoder-based features?
>
> **Response 3:**
>
> We appreciate the reviewer’s question about the choice of auto-encoder-based features for unsupervised representation learning. As we mentioned in our response to **Weakness 3**, auto-encoder-based representation learning was chosen because **it is effective when the manifold assumption holds**, which is the most suitable for two-sample testing data.
>
> On the other hand, other popular unsupervised representation learning techniques, such as SimCLR [1] and SwAV [2], which still utilize the point-wise semi-supervised learning techniques such as label propagation or sample augmentation. These approaches are not directly viable in the field of two-sample testing due to the violation of the smoothness assumption and cluster assumption in this context. Moreover, if we preserve the sample index of two samples $S_P$ and $S_Q$, then conduct a representation learning to learn a feature extractor that can best separate two samples, we will face a problem in the hypothesis testing is that the type-I error will be out of control, since using unlabeled testing data to learning DRs will always make the probability of successfully discriminating two samples from the same distribution be higher than the significance level $\alpha$.
>
> We hope this clarifies why auto-encoder-based representation learning was selected as the most suitable method for our framework. Moreover, we will add the above discussion into the revised paper to help readers to better understand the motivation why we choose AE-based representation learning rather than unsupervised contrastive representation learning.
>
> [1] Chen T, Kornblith S, Norouzi M, Hinton G. A simple framework for contrastive learning of visual representations. InInternational conference on machine learning 2020 Nov 21 (pp. 1597-1607). PMLR.
>
> [2] Caron M, Misra I, Mairal J, Goyal P, Bojanowski P, Joulin A. Unsupervised learning of visual features by contrasting cluster assignments. Advances in neural information processing systems. 2020;33:9912-24.
>
> **We would be delighted to engage in further discussions to ensure your concerns are fully addressed.**

---

> > ### Author Response · Authors · 2024-11-26
> >
> > Dear Reviewer Y9aR,
> >
> > We appreciate the time and effort that you have dedicated to reviewing our manuscript.
> >
> > We have carefully addressed all your queries. Could you kindly spare a moment to review our responses?
> >
> > Have our responses addressed your major concerns?
> >
> > If there is anything unclear, we will address it further. We look forward to your feedback.
> >
> > Best regards,
> >
> > The authors of Submission 6635

---

> > > ### Author Response · Authors · 2024-11-26
> > >
> > > Dear Reviewer Y9aR,
> > >
> > > Thank you again for your time and effort in reviewing our paper.
> > >
> > > We have carefully addressed all your comments and prepared detailed responses. Could you kindly review them at your earliest convenience?
> > >
> > > As the discussion period is coming to a close, we wanted to check if there are any remaining concerns preventing you from adjusting your score. If there is any additional clarification we can provide, please do not hesitate to let us know, and we will address it promptly.
> > >
> > > Best regards,
> > >
> > > Authors of Submission 6635

---

> ### Author Response · Authors · 2024-11-29
>
> Dear Reviewer Y9aR,
>
> Thank you for your valuable feedback and constructive comments on our submission.
>
> Since we have already solve all the concerns that has been proposed, we would like to ask if you have any additional concerns or questions about the paper that we can address. If there are specific points that remain unclear, please feel free to point them out—we are more than willing to discuss and provide further explanations.
>
> If all your concerns have been addressed satisfactorily, we would sincerely appreciate it if you could consider revisiting your scores and generously increasing them, if appropriate, to reflect the clarified contributions of the paper.
>
> Thank you once again for your thoughtful review and for helping us improve our work.
>
> Best regards,
>
> Authors of Submission 6635

---

> > ### Author Response · Authors · 2024-11-30
> >
> > Dear Reviewer Y9aR,
> >
> > We hope this message finds you well. As the closing date for the review process is approaching, we wanted to follow up to ensure that all of your concerns regarding our paper have been addressed. If there are any remaining questions or issues you would like us to clarify, we would be more than happy to discuss them further.
> >
> > If all your concerns have been resolved, we kindly ask if you could consider revisiting your scores and updating them to reflect the clarified contributions of our work. Your constructive feedback has been invaluable, and we sincerely appreciate the time and effort you have devoted to reviewing our paper.
> >
> > Thank you once again for your thoughtful evaluations, and we look forward to hearing from you.
> >
> > Best regards,
> >
> > Authors of Submission 6635

---

> > > ### Author Response · Authors · 2024-12-01
> > >
> > > Dear Reviewer Y9aR,
> > >
> > > We hope this message finds you well. As the closing date for the review process is approaching, we wanted to follow up to ensure that all of your concerns regarding our paper have been addressed. If there are any remaining questions or issues you would like us to clarify, we would be more than happy to discuss them further.
> > >
> > > If all your concerns have been resolved, we kindly ask if you could consider revisiting your scores and updating them to reflect the clarified contributions of our work. Your constructive feedback has been invaluable, and we sincerely appreciate the time and effort you have devoted to reviewing our paper.
> > >
> > > Thank you once again for your thoughtful evaluations, and we look forward to hearing from you.
> > >
> > > Best regards,
> > >
> > > Authors of Submission 6635

---

> > > > ### Comment · Reviewer_Y9aR · 2024-12-02
> > > > **Response to the Authors**
> > > >
> > > > I appreciate the authors for providing a detailed and thorough rebuttal. After carefully reviewing it, I find that most of my initial concerns and questions have been addressed. This work is valuable in informing the community about the effectiveness of autoencoder-based SSL for two-sample testing, while highlighting that a naive application of other SSL methods may not yield similar benefits. While I am inclined to accept this work, I maintain my original rating due to the limited novelty of the method.

---

> ### Author Response · Authors · 2024-12-02
> **Thanks for recognizing our main contributions to the community!**
>
> Dear Reviewer Y9aR,
>
> **Many thanks for recognizing our contributions to the field.** We think you have concluded our contributions clearly:
>
> >This work is valuable in informing the community about the effectiveness of autoencoder-based SSL for two-sample testing, while highlighting that a naive application of other SSL methods may not yield similar benefits.
>
> **Novelty concern.** As for the novelty of our method, we want to emphasise that our main contribution is to investigate how SSL works for two-sample testing problem at **its first place**. It is always difficult **to be the first work** to pave a new perspective to understand an important problem, and we did a thorough investigation on this point in the current paper.
>
> Best regards,
>
> Authors of Submission 6635

---

### Author Response · Authors · 2024-11-28

We sincerely thank the reviewers for their insightful comments, which have greatly contributed to improving our work. We have carefully revised our paper based on the discussions and feedback provided.

Thanks for Reviewer **Y9aR**
* We add the extra proof of why it is obvious that a max objective can be equivalent to the final min objective.
* We revise the repeated / emphasized part in Section 3. We now use the **research questions** to lead the presentation of our work.

Thanks for Reviewer **Y9aR** and **47NZ**
* We add the extra explanation about the motivation why selecting the AE-based representation learning into our proposed method.

Thanks for Reviewer **Y9aR** and **4szV**
* We emphasize the original definition of the SSL, and why our proposed method is satisfied with that definition.

Thanks for Reviewer **4szV**
* We make the notation N clear defined in the description of figure, and add an extra notation M to be different in the description of table.
* We restate the implementation details of C2ST (e.g. model architecture, train-test spliting proportion) from the referenced paper.

---

### Meta-Review · Area_Chair_sa4C · 2024-12-17

**Metareview:**

This paper proposes a novel perspective for two-sample testing based on semi-supervised learning that utilize both labeled and unlabeled samples called SSL-C2T.  A straightforward implementation of SSL-C2T does not work, due to the strong assumptions of popular semi-supervised learning algorithms such as the manifold, cluster, or smoothness assumption. These assumptions do not fit the problem of two-sample testing, because we usually try to tackle datasets where the two samples are similar. The paper further proposes a two-step approach to first learn inherent representations using all data, then fine-tune the representations with labeled data. Experimental results and theoretical analysis are provided.

Some reviewers found the motivation and problem definition clear, and some reviewers appreciated the novelty of applying semi-supervised learning to two-sample testing. There are both theoretical justification and empirical investigation in the paper.

On the other hand, there were some concerns regarding the "methodological novelty" and that the paper still needs to demonstrate the practicality (good performance) in experiments, such as using datasets other than the vision domain. There were some questions about other problem settings such as out-of-distribution setup and positive-unlabeled classification setup, as well as some clarification questions about the theoretical analysis.

Most of the concerns and questions were addressed well during the rebuttal period. The main concern that remained among the reviewers is the contribution w.r.t. "methodological novelty" and "practicality". The authors emphasized the "conceptual-level contribution" of revisiting the two-sample testing as a SSL problem and that the aim is not to propose a practical SOTA method for this, but the final reviewer ratings were 3 borderline scores (6,6,6) and two lower scores (3,3) with an average of 4.8. Conceptual novelty can be a significant contribution, but the paper may need to further emphasize this point, so that the reviewers can appreciate the conceptual contribution of the paper. A suggestion is to expand some discussions about potential future directions so that the reader (or reviewer) can see hints of a lot of follow-up work from this paper. This will make it easier to appreciate the conceptual novelty and novel perspective that the paper provides, and put less weight on immediate impact.

**Additional Comments On Reviewer Discussion:**

Two reviewers provided a score of 3. Both of them engaged with the authors during the rebuttal phase but did not change their final scores.

---

### Decision · Program_Chairs · 2025-01-22

Reject